# MANAGING TEMPORAL RESOLUTION IN CONTINUOUS VALUE ESTIMATION: A FUNDAMENTAL TRADE-OFF

## ABSTRACT

A default assumption in reinforcement learning and optimal control is that experience arrives at discrete time points on a fixed clock cycle. Many applications, however, involve continuous systems where the time discretization is not fixed but instead can be managed by a learning algorithm. By analyzing Monte-Carlo value estimation for LQR systems in both finite-horizon and infinite-horizon settings, we uncover a fundamental trade-off between approximation and statistical error in value estimation. Importantly, these two errors behave differently with respect to time discretization, which implies that there is an optimal choice for the temporal resolution that depends on the data budget. These findings show how adapting the temporal resolution can provably improve value estimation quality in LQR systems from finite data. Empirically, we demonstrate the trade-off in numerical simulations of LQR instances and several non-linear environments.

## 1 INTRODUCTION

In many real-world applications of control and reinforcement learning, the underlying system evolves continuously in time. For instance, a physical system such as a robot is naturally modeled as a stochastic dynamical system. In practice, however, sensor measurements are usually captured at discrete time intervals, and the practitioner must make a decision about how to discretize the time dimension, i.e. choosing a sampling frequency or a measurement *step-size*. A common belief is that a finer time discretization always leads to better estimation of the system properties and the control cost or the reward in reinforcement learning. As we show, this is only true with an unlimited data budget. In practice there are always limitations on how much data can be collected, stored and processed. Consider for example the task of episodic policy evaluation with a finite data budget. A higher temporal resolution means that *more* data is collected within *fewer* episodes. This inevitably leads to the question on how to *optimally* choose the time discretization for the task at hand.

The practitioner therefore faces a fundamental trade-off: using a finer temporal resolution leads to better approximation of the continuous-time system from discrete measurements, but the consequence of collecting denser data along fewer trajectories leads to larger estimation variance with respect to stochasticity in the system. This is indeed true for any system with stochastic dynamics, even if the learner has access to *exact* (noiseless) measurements of the system's state. In this paper, we show that data efficiency can be significantly improved by leveraging a precise understanding of the trade-off between approximation error and statistical estimation error in long term value estimation — two factors that react differently to the level of temporal discretization.

The main contributions of this work are twofold. First, we consider the simplest and canonical case of Monte-Carlo value estimation in a Langevin dynamical system (linear dynamics perturbed by a Wiener process) with quadratic instantaneous costs. Although the setup is specialized, it is simple enough such that we can obtain analytical expressions of the least-squares error that *exactly characterize the approximation-estimation trade-off* with respect to the step-size parameter. Second, we present a *numerical study* that illustrates and confirms the trade-off in both *linear and non-linear* systems, including several MuJoCo control environments. Our findings imply that practitioners should pay attention to carefully choosing the step-size parameter of the estimation to obtain the most accurate results possible.

## 1.1 RELATED WORK

There is a sizable literature on reinforcement learning in continuous-time systems (e.g. Doya, 2000; Lee & Sutton, 2021; Lewis et al., 2012; Bahl et al., 2020; Kim et al., 2021; Yildiz et al., 2021). These previous works have largely focused on deterministic dynamics, and do not investigate trade-offs in temporal discretization. A smaller body of work has considered learning continuous-time control under stochastic (Baird, 1994; Bradtke & Duff, 1994; Munos & Bourgine, 1997; Munos, 2006), or bounded (Lutter et al., 2021) perturbations, but with a focus on making standard learning methods more robust to small time scales (Tallec et al., 2019), again without explicitly managing the temporal discretization level. There have also been works that characterize the effects of temporal truncation in infinite horizon problems (Jiang et al., 2016; Droge & Egerstedt, 2011). Despite these prevailing topics in the literature, we find that managing temporal discretization offers substantial improvements not captured by these previous studies.

The LQR setting is a standard framework in control theory and it gives rise to a fundamental optimal control problem (Lindquist, 1990), which has proven itself to be a challenging scenario for Reinforcement Learning algorithms (Tu & Recht, 2019; Krauth et al., 2019). The stochastic LQR considers linear systems driven by additive Gaussian noise with a quadratic form for the cost, which is sought to be minimised by means of a feedback controller. Although it is a well-understood scenario and a closed form of the optimal controller is known thanks to the separation principle (Georgiou & Lindquist, 2013), only recently the statistical properties of the long-term cost have been investigated (Bijl et al., 2016). The work in this paper also closely related to the now sizable literature on reinforcement learning in LQR systems (Bradtke, 1992; Krauth et al., 2019; Tu & Recht, 2018; Dean et al., 2020; Tu & Recht, 2019; Dean et al., 2018; Fazel et al., 2018; Gu et al., 2016). These existing works uniformly focused on the discrete time setting, although the benefits of managing spatial rather than temporal discretization has been considered (Sinclair et al., 2019; Cao & Krishnamurthy, 2020). Wang et al. (2020) studied the continuous-time LQR setting but it focused on the exploration problem rather than the temporal discretization.

There is compelling empirical evidence that managing temporal resolution, typically via action persistence (Lakshminarayanan et al., 2017; Sharma et al., 2017; Huang et al., 2019; Huang & Zhu, 2020; Dabney et al., 2021; Park et al., 2021), can greatly improve learning performance. Even grid worlds (Sutton & Barto, 2018) can be seen as leveraging a form of action persistence, where a coarse spatial discretization is imposed on an otherwise continuous two dimensional navigation problem to improve learning efficiency. These empirical findings have recently been supported by an initial theoretical analysis (Metelli et al., 2020) that shows temporal discretization plays a role in determining the effectiveness of fitted Q-iteration. The analysis by Metelli et al. (2020) does not consider fully continuous systems, but rather remains anchored in a base level discretization and only provides worst-case upper bounds that do not necessarily capture the detailed trade-offs one faces in practice.

Choosing the temporal resolution can also be understood as a non-linear *experimental design* problem (Chaloner & Verdinelli, 1995; Ford et al., 1989). By choosing the time discretization, the experimenter determines how to allocate measurements for a given data budget. What is peculiar to our objective is that any fixed design has a constant approximation error (bias) that persists even when the number of data points becomes infinite. At the same time, the bias can also be managed by scarifying estimation error (variance). Optimal designs that consider the bias-variance trade-off jointly have been studied previously (e.g. Bardow, 2008; Mutny et al., 2020; Mutný & Krause, 2022).

## 2 POLICY EVALUATION IN CONTINUOUS LINEAR QUADRATIC SYSTEMS

In the classical continuous-time linear quadratic regulator (LQR), a state variable $X(t) \in \mathbb{R}^n$ evolves over time $t \geq 0$ according to the following equation:

$$\mathrm{d}X(t) = \mathbf{A}X(t)\,\mathrm{d}t + \mathbf{B}U(t)\,\mathrm{d}t + \sigma dW(t). \tag{1}$$

The dynamical model is fully specified by the matrices $\mathbf{A} \in \mathbb{R}^{n \times n}$, $\mathbf{B} \in \mathbb{R}^{n \times p}$ and the diffusion coefficient $\sigma$. The control input is $U(\cdot) \in \mathbb{R}^p$ is given by a fixed policy, and $W(t)$ is a Wiener process. The state variable $X(t)$ is fully observed. For simplicity, we assume that the dynamics start at $X(0) = \overrightarrow{0} \in \mathbb{R}^n$ (c.f. Abbasi-Yadkori & Szepesvári, 2011; Dean et al., 2020).

The expected quadratic cost $J$ is defined for positive semi-definite, symmetric matrices $\mathbf{Q} \in \mathbb{R}^{n \times n}$ and $\mathbf{R} \in \mathbb{R}^{p \times p}$, a *system horizon* $0 < \tau \leq \infty$ and a discount factor $\gamma \in (0, 1]$:

$$J_\tau = \int_0^\tau \gamma^t \left[ X(t)^\top \mathbf{Q} X(t) + U(t)^\top \mathbf{R} U(t) \right] dt \tag{2}$$

In the following we consider the class of controllers given by static feedback of the state, i.e.:

$$U(t) = KX(t) \tag{3}$$

where $K \in \mathbb{R}^{p \times n}$ is the static control matrix yielding the control input. It is well known that in infinite horizon systems with discounting, the optimal control is of the form Eq. (3). The specific choice of policy plays no particular role in what follows, therefore we reduce the LQR in Eq. (1) further to a linear stochastic dynamical system described by a Langevin equation. Using the definitions $A := \mathbf{A} + \mathbf{B}K$ and $Q := \mathbf{Q} + K^\top RK$, we express both the state dynamics and the cost in a more compact form:

$$\mathrm{d}X(t) = AX(t) + \sigma\xi(t), \qquad J_\tau = \int_0^\tau \gamma^t X(t)^\top QX(t) \, dt \tag{4}$$

The expected cost $V_\tau$ is the expectation of the cost w.r.t. the Wiener process, $V_\tau = \mathbb{E}[J_\tau]$.

Equation (4) is what we analyze in the following. From now on, we explicitly distinguish the *finite-horizon setting* where $\tau < \infty$, $\gamma \leq 1$ and the cost is $V_\tau$, and the *infinite-horizon setting* where $\tau = \infty$, $\gamma < 1$ and the cost is $V_\infty$.

**Monte-Carlo Policy Evaluation** Our main objective of *policy evaluation* is to estimate the expected cost from discrete-time observations. To this end, we choose a uniform discretization of the interval $[0, T]$ with increment $h$ resulting in $N = T/h$ time points $t_k := kh$ for $k \in \{0, 1, \ldots, N\}$. Here, the *estimation horizon* $T$, such that $T < \infty$ and $T \leq \tau$, is chosen by the practitioner (for simplicity assume that $T/h$ is an integer). With the $N$ points sampled from one trajectory, a standard way to approximate the integral in Eq. (4) is with the *Riemann sum estimator*

$$\hat{J}(h) = \sum_{k=0}^{N-1} \gamma^{t_k} h X(t_k)^\top QX(t_k). \tag{5}$$

To estimate $V_\tau$, we average $M$ independent trajectories with cost estimates $\hat{J}_1, \ldots \hat{J}_M$ to obtain the *Monte-Carlo estimator*:

$$\hat{V}_M(h) = \frac{1}{M} \sum_{i=1}^M \hat{J}_i(h) = \frac{1}{M} \sum_{i=1}^M \sum_{k=0}^{N-1} \gamma^{t_k} h X(t_k)^\top QX(t_k) \tag{6}$$

Our main objective is to understand the mean-squared error of the Monte-Carlo estimator for a fixed system (specified by $A$, $\sigma$ and $Q$), with the goal to inform an optimal choice of the step-size parameter $h$ for a *fixed data budget $B = M \cdot N$*.

Note that one degree of freedom remains in choosing $M$ and $N$. For simplicity, we require that in the finite-horizon setting, the estimation grid is chosen to cover the full episode $[0, \tau]$ which leads to the constraint $T = \tau = N \cdot h$. We write the least-squares error-surface as a function of $h$ and $B$:

$$\mathrm{MSE}_T(h, B) = \mathbb{E}\left[(\hat{V}_M(h) - V_T)^2\right] \tag{7}$$

In the infinite horizon setting, i.e. $\tau = \infty$, the *estimation horizon $T$* is a free variable chosen by the experimenter that determines the number of trajectories through $M = \frac{B}{N} = \frac{Bh}{T}$. The mean-squared error for the infinite horizon setting is given as a function of $h$, $B$, and $T$:

$$\mathrm{MSE}_\infty(h, B, T) = \mathbb{E}\left[(\hat{V}_M(h) - V_\infty)^2\right]. \tag{8}$$

## 3 CHARACTERIZING THE MEAN-SQUARED ERROR

In the following our goal is to characterize the least-squares error of the Monte-Carlo estimator as a function of the step size $h$ and the total data budget $B$ (and the estimation horizon $T$ in the infinite horizon setting). Our results uncover a fundamental trade-off for choosing an *optimal* step size that leads to a minimal least-squares error.

**One-Dimensional Langevin Process**    To simplify the exposition while preserving the main ideas, we will first present the results for the 1-dimensional case. The analysis for the vector case exhibits the same quantitative behavior but is significantly more involved. To distinguish the 1-dimensional from the $d$-dimensional setting described in Eq. (4), we use lower-case symbols. Let $x(t) \in \mathbb{R}$ be the scalar state variable that evolves according to following the Langevin equation:

$$dx(t) = ax(t)dt + \sigma dw(t). \tag{9}$$

Here, $a \in \mathbb{R}$ is the drift coefficient and $w(t)$ is a Wiener process with scale parameter $\sigma > 0$. We assume that $a \leq 0$, i.e. the system is stable (or marginally stable).

The realized sample path in episode $i = 1, \ldots, M$ is $x_i(t)$ (with starting state $x(0) = 0$) and $t \in [0, T]$. The expected cost is

$$V_\tau = \mathbb{E}\Big[ \int_0^\tau \gamma^t r_i^2(t) dt \Big] = \int_0^\tau \gamma^t q \mathbb{E}\big[x_i^2(t)\big] dt, \tag{10}$$

where $r_i(t) = qx_i(t)^2$ is the quadratic cost function for a fixed $q > 0$. The Riemann sum that approximates the cost realized in episode $i \in [M]$ becomes $\hat{J}_i(h) = \sum_{k=0}^{N-1} hqx_i^2(kh)$. Given data from $M$ episodes, the Monte-Carlo estimator is $\hat{V}_M(h) = \frac{1}{M} \sum_{i=1}^{M} \hat{J}_i(h)$. Since the square of the cost parameter $q^2$ factors out of the mean-squared error in Eq. (10), we set $q = 1$ in what follows.

## 3.1    Finite-Horizon

Recall that in the finite-horizon setting we set the system horizon $\tau$ and estimation horizon $T$ to be the same. This implies that the estimation grid covers the full episode, i.e. $hN = T = \tau$. Perhaps surprisingly, the mean-squared error of the Riemann estimator for the Langevin system (9) can be computed in closed form. The result takes its simplest form in the finite-horizon, undiscounted setting where $\gamma = 1$ and $\tau < \infty$. This result is summarized in the next theorem.

**Theorem 1** (Finite-horizon, undiscounted MSE). *In the finite-horizon, undiscounted setting, the mean-squared error of the Monte-Carlo estimator is*

$$\mathrm{MSE}_T(h, B) = E_1(h, T, a) + \frac{E_2(h, T, a)}{B},$$

*where*

$$E_1(h, T, a) = \frac{\sigma^4 \left(-2ah + e^{2ah} - 1\right)^2 \left(e^{2aT} - 1\right)^2}{16a^4 \left(e^{2ah} - 1\right)^2},$$

$$E_2(h, T, a) = \frac{\sigma^4 T \left[h \left(e^{2aT} - 1\right) \left(4e^{2ah} + e^{2aT} + 1\right) - \left(e^{2ah} - 1\right) \left(e^{2ah} + 4e^{2aT} + 1\right) T\right]}{2a^2 \left(e^{2ah} - 1\right)^2}.$$

The proof involves computing the closed-form expressions for the second and forth moments of the random trajectories $x_i(t)$ and is provided in Appendices B and C.1. While perhaps daunting at first sight, the beauty of the result is that it *exactly* characterizes the error surface as a function of the step size $h$ and the budget $B$.

In principle, for any fixed $B$, we can optimize $h$ to minimize the mean-squared error by searching over possible step-sizes $h_m = T/m$ for $m = 1, \ldots, B$, provided knowledge of the system parameters $a$, $\sigma$ and fixed horizon $T$. On the other hand, the practical scope of this procedure is somewhat limited. On the upside, as we show next, the underlying trade-offs can be characterized and understood closely in several different regimes. In Section 4, we show through numerical experiments how these insights translate into simulations of linear and non-linear systems.

In the case of marginal stability ($a = 0$), a simpler form of the MSE emerges that is easier to interpret. Taking the limit $a \to 0$ of the previous expression gives the following result:

**Corollary 1** (MSE for marginally stable system). *Assume a marginally stable system, $a = 0$. Then the mean-squared error of the Monte-Carlo estimator is*

$$\mathrm{MSE}_T(h, B) = \frac{\sigma^4 T^2}{4} \cdot h^2 + \frac{\sigma^4 T^5}{3} \cdot \frac{1}{hB} + \frac{\sigma^4 T^2(-2T^2 + 2hT - h^2)}{3B}.$$

The first part of the expression can be understood as a Riemann sum *approximation error* that is controlled by the $h^2$ term. The second part corresponds to the *variance term* that decreases with the number of episodes as $\frac{1}{M} = \frac{T}{Bh}$. The remaining terms are of lower order terms for small $h$ and large $B$. For a fixed data budget $B$, the step size $h$ can be chosen to balance these two terms (up to lower order terms in $1/B$):

$$h^*(B) := \underset{h>0}{\arg\min} \, \text{MSE}_T(h, B) \approx T \left( \frac{2}{3B} \right)^{1/3}. \tag{11}$$

From this, we can compute the optimal number of episodes $M^* \approx \frac{Bh}{T} = \left( \frac{2}{3} \right)^{1/3} B^{2/3}$. We remark that under the assumption $B \gg 1$, we also obtain that $M^* \gg 1$. This is in agreement with the implicit requirement that $h$ is big enough to consider at least one whole trajectory, i.e. $h > T/B$. Consequently, the mean-squared error for the optimal choice of $h$ (up to lower order terms in $1/B$):

$$\text{MSE}_T(h^*, B) \approx 3 \, (3/2)^{1/3} \, \sigma^4 T^4 B^{-2/3}.$$

In other words, the optimal error rate as a function of the data budget is $\mathcal{O}(B^{-2/3})$.

We can further obtain a similar form for $h^*$ for the general case where $a \leq 0$.

**Corollary 2** (Optimal step size). *For $B \gg 1$, the optimal step-size (up to lower order terms in $1/B$)*

$$h^*(B) \approx \left( -\frac{T \left( 4aT - e^{4aT} + e^{2aT}(8aT - 4) + 5 \right)}{a^2 (e^{2aT} - 1)^2} \right)^{1/3} B^{-1/3}.$$

*Moreover,* $\text{MSE}_T(h^*, B) \leq \mathcal{O}(B^{-2/3})$.

The proof is provided in Appendix C.2 where we also include a more precise expression of $h^*$.

**Discounted Cost** Adding discounting ($\gamma < 1$) in the finite-horizon setting does not fundamentally change the results but makes it more involved (details shown in Appendix C.3).

**Vector Case** Also for the vector case ($n > 1$) it is possible to exactly characterise the mean-squared error of the Monte-Carlo estimator for the Langevin system in Eq. (9). The closed-form computations will however require to assume that the matrix $A$ governing the behaviour of the system is diagonalisable, and stable. The latter is a rather mild assumption, as it is sufficient for the system in Eq. (1) to be controllable to ensure satisfiability of this condition. Controllability in fact translates into the possibility of freely adjusting the eigenvalues of the closed-loop matrix $A$ through the choice of the controller $K$. This means that it is always possible to choose eigenvalues to be distinct from each other, so that $A$ is diagonalisable. The explicit form of the mean-squared error, although actually computable, is given by a long formula which is not easy to interpret, and is therefore deferred to Appendix D. The following theorem summarizes the result for the vector case in the form of a Taylor expansion for small $h$ and large $B$.

**Theorem 2** (Mean-squared error - vector case). *Assume $A$ is diagonalisable, with eigenvalues $\lambda_1, \ldots, \lambda_n$. The mean-squared error of the Monte-Carlo estimator in the finite-horizon, undiscounted setting, is*

$$\text{MSE}_T(h, B) = E_1(h, T, \lambda_1, \ldots, \lambda_n) + \frac{E_2(h, T, \lambda_1, \ldots, \lambda_n)}{B} \tag{12}$$

*where*

$$E_1(h, T, \lambda_1, \ldots, \lambda_n) = \left( \overline{C}_1 + C_1(\lambda_1, \ldots, \lambda_n) \mathcal{O}(T) \right) \sigma^4 T^2 h^2 + \mathcal{O}(h^3) \tag{13}$$

$$\frac{E_2(h, T, \lambda_1, \ldots, \lambda_n)}{B} = \left( \overline{C}_2 + C_2(\lambda_1, \ldots, \lambda_n) \mathcal{O}(T) \right) \sigma^4 \frac{T^5}{hB} + \mathcal{O}(1/B) \tag{14}$$

The proof with the exact derivation of the constants $\overline{C}_1, C_1(\lambda_1, \ldots, \lambda_n), \overline{C}_2, C_2(\lambda_1, \ldots, \lambda_n)$ can be found in Appendix D.1. Note that the terms composing the MSE are very similar to the ones obtained in the scalar analysis. Indeed by comparing them with the expressions in Eq. (28) and Eq. (29)

(in Appendix C.2), the expression has the same order for $h, B$ and $T$. The only difference is that in the vector case, cumbersome eigenvalue-dependent constants are involved, whereas in the scalar case the result can more easily be expressed in terms of the system parameter $a$.

Since the optimal choice for $h$ is given by balancing the trade-off between the two terms above, $E_1$ for the approximation error and $E_2$ for the variance term, its expression is analogous to the scalar case, as shown by the following corollary.

**Corollary 3** (Optimal step size - vector case). *Under the assumption that $B \gg 1$, the optimal step-size for the vector case is given by*

$$h^*(B) = \left( \frac{\overline{C}_1 + C_1(\lambda_1, \ldots, \lambda_n) \mathcal{O}(T)}{\overline{C}_2(\lambda_1, \ldots, \lambda_n) + C_2(\lambda_1, \ldots, \lambda_n) \mathcal{O}(T)} \right)^{1/3} TB^{-1/3} + o(B^{-1/3}) \quad (15)$$

The constants in Corollary 3 are clearly the same as in Theorem 2.

General bounds that hold for the case of a vector Langevin process with a stable matrix $A$ are provided in Appendix D.3. These results show that the mean-squared error lies in between two expressions with the same order in $h$ and $B$, whose difference depends only on $T$, and the eigenvalues of the matrix. Both the lower and upper bounds are convex functions of $h$, narrowing down the behaviour of the step size in this general case. In particular, the lower bound can always be expressed in terms of the mean-squared error for the scalar case, emphasizing the importance of examining this special case. Although the convex behaviour is only proven for the case of a Langevin system, our experimental results (Section 4) exhibit a similar trade-off for general nonlinear stochastic systems.

From the present analysis, it is possible to derive guidelines on how to set the step-size even for the case of nonlinear and unknown dynamics. Although the sharp order in B for the optimal step-size holds for the case of linear dynamics only, we empirically show in Section 4 that a similar trade-off carries on to nonlinear dynamics, and $h = cTB^{-1/3}$ is a solid choice for the more general setting. While the constant $c$ depends on the controlled dynamics (therefore on both the free dynamics and the policy), $c$ could be estimated with a small budget, in order to properly scale the value of $h$ for a large-scale experiment. This approach does not require the knowledge of the dynamics beforehand, nonetheless it provides a systematic way of setting the step size $h$ for any given scenario.

## 3.2 INFINITE-HORIZON SETTING

The main characteristic of the finite-horizon setting is the trade-off between approximation and estimation error. Recall that in the infinite-horizon setting ($\tau = \infty$), the estimation horizon $T < \infty$ becomes a free variable that is chosen by the experimenter to define the measurement range $[0, T]$. Consequently the mean-squared error of the Monte-Carlo estimator suffers an additional *truncation error* from using a finite Riemann sum with $N = T/h$ terms as an approximation to the infinite integral that defines the cost $V_\infty$.

More precisely, we decompose the expected cost $V_\infty = V_T + V_{T,\infty}$, where $V_T = \int_0^T \gamma^t \mathbb{E}[x^2(t)]dt$ as before, and

$$V_{T,\infty} = \int_T^\infty \gamma^t \mathbb{E}\left[x^2(t)\right] \, \mathrm{d}t = \frac{\sigma^2 \gamma^T}{2a} \left( \frac{1}{\log(\gamma)} - \frac{e^{2aT}}{\log(\gamma) + 2a} \right). \quad (16)$$

The integral is a direct calculation based on Lemma 1 in Appendix B. Thus the mean-squared error becomes

$$\mathrm{MSE}_\infty(h, B, T) = \mathbb{E}\left[(\hat{V}_M(h) - V)^2\right] = \mathrm{MSE}_T(h, B) - 2\mathbb{E}\left[\hat{V}_M(h) - V_T\right] V_{T,\infty} + V_{T,\infty}^2, \quad (17)$$

where $\mathrm{MSE}_T(h, B) = \mathbb{E}\left[(\hat{V}_M(h) - V_T)^2\right]$ is the mean-squared error of discounted finite-horizon setting. Note that the term $V_{T,\infty}^2$ is neither controlled by a small step-size $h$ nor by a large data budget $B$, hence results in the truncation error from finite estimation. Fortunately, the geometric discounting ensures that $V_{T,\infty}^2 = \mathcal{O}(\gamma^{2T})$, which is not unexpected given that the term constitutes the tail of the geometric integral. In particular, setting $T = c \cdot \log(B)/\log(1/\gamma)$ for large enough $c > 1$ suffices to ensure that the truncation error is below the estimation variance.

We summarize the result in the next theorem.

**Theorem 3** (Infinite-horizon, discounted MSE). *In the infinite-horizon, discounted setting, the mean-squared error of the Monte-Carlo estimator is*

$$\text{MSE}_\infty(h, B, T) = \sigma^4 \, T \, C(a, \gamma) \cdot \frac{1}{hB} + \frac{\sigma^4}{144} \cdot h^4 + \mathcal{O}(h^5) + \mathcal{O}(B^{-1}) \tag{18}$$

*where we let* $C(a, \gamma) = \frac{1}{\log(\gamma)(a+\log(\gamma))(2a+\log(\gamma))^2}$ *and assume that* $\gamma^T = o(h^4)$.

It follows that the optimal choice for the step-size is $h^*(B, T) \approx (36 \, T \, C(a, \gamma)/B)^{1/5}$. The minimal least-squares error is $\text{MSE}_\infty(h^*, T, B) \leq \mathcal{O}\big((T \, C(a, \gamma)/B)^{4/5} + \gamma^{2T}\big)$.

Lastly, we remark that if $\gamma^T$ is treated as a constant, the cross term $\mathbb{E}\big[\hat{V}_M(h) - V_T\big] V_{T,\infty}$ in Eq. (17) introduces a dependence of order $\mathcal{O}(h\gamma^{2T})$ to the mean-squared error. In this case, the overall trade-off becomes $\text{MSE}_\infty(h, B, T) \approx \mathcal{O}\big(1/(hB) + \gamma^{2T}(1+h)\big)$, and the optimal step-size is $h^* \approx B^{-1/2}$.

**Vector Case**   As before, the mean-squared error for the vector case can be explicitly computed in closed-form assuming that $A$ is diagonalizable. The result reflects the same behaviour as in the scalar case. Conveniently, the MSE in Theorem 3 has been expressed with sharp terms in $h$ and $B$, while confining the dependence on the system parameter $a$ within the constant $C$, and the impact of higher-order terms in $T$ within $V_{T,\infty}$. This allows us to state the same result for the vector case, in which the constant will now depend on the eigenvalues of the matrix $A$, as well as the discount factor $\gamma$. These are provided in full detail in Appendix D.2.

**Corollary 4.** *For $A$ diagonalisable, with eigenvalues given by $\lambda_1, \ldots, \lambda_n$, the mean-squared error of the Monte-Carlo estimator in the infinite-horizon, discounted setting is*

$$\text{MSE}_\infty(h, B, T) = C_3(\lambda_1, \ldots, \lambda_n, \gamma) \, \sigma^4 \frac{T}{hB} + \frac{\sigma^4}{144} h^4 + \mathcal{O}(h^5) + \mathcal{O}(B^{-1}), \tag{19}$$

*under the assumption that* $\gamma^T = o(h^4)$.

The different terms in Corollary 4 correspond to the estimation error, the approximation error and the truncation error as in the scalar case. The optimal step size choice exhibits the same dependence on $T$ and $B$ as in the scalar case, but with a different constant depending on the eigenvalues. Lastly, the general case for Langevin processes with a stable matrix $A$ is discussed in Appendix D.3.

## 4   FROM LINEAR TO NON-LINEAR SYSTEMS: A NUMERICAL STUDY

The trade-off identified in our analysis suggests that there exists an optimal choice of the temporal resolution in policy evaluation. Our next goal is to verify the trade-off in several simulated dynamical systems. While our analysis assumes a linear transition and quadratic cost, we empirically demonstrate that such a trade-off also exists in nonlinear systems. For our experimental setup, we choose simple linear quadratic systems mirroring the setup of Section 2, as well as several standard benchmarks from OpenAI Gym (Brockman et al., 2016) and MuJoCo (Todorov et al., 2012). Our findings confirm the theoretical results and highlight the importance of choosing an appropriate step-size for policy evaluation.

### 4.1   LINEAR QUADRATIC SYSTEMS

We first run numerical experiments on the Langevin dynamical systems to demonstrate the trade-off analyzed in the previous section, the results of this experiment are shown in Fig. 1. For all the systems, we fix the parameters $\sigma^2 = 1$ and $Q = I$. The lines in the plot represent the sample mean $(\hat{V}_M(h) - V)^2$ and the shading represent standard error of our sample means. Each data point was averaged over 50 independent runs in the scalar case and 40 in the vector case. We observe a clear trade-off in all the Fig. 1 plots. Fig. 1(a) shows the MSE in an one-dimensional system with $T = 8$ and $a = -1$. The ground truth $V$ is calculated analytically by using Eq. 27. The figure illustrates how the error changes as we vary the data budget, $B = \{2^{12}, 2^{13}, 2^{14}, 2^{15}, 2^{16}\}$, and also illustrates the improvement that can be obtained by increasing the budget. As we increase $B$, both the error and and the optimal step size, $h^*$, decrease. This result aligns with the analysis shown in Theorem 1

and Corollary 2. The objective is minimized when $h$ is chosen appropriately. Fig. 1(b) and 1(c) show the experimental results for both undiscounted finite horizon and discounted infinite horizon multi-dimensional systems. Calculating the ground truth $V$ for our multi-dimensional systems is more involved than in our scalar system. To calculate $V$ for the undiscounted finite horizon system, we numerically solve the Riccati Differential Equation using backwards induction as is standard practice. To calculate $V$ for the discounted infinite horizon system we solve the Lyapunov equation using a standard solver. Note in our experiments the dimension $n = 3$. We fix all parameters and run our experiments on the system $A = cI_3$ where $c \in \{-0.2, -0.5, -1, -2, -4\}$, producing stable systems with different eigenvalues. Results in both plots suggest that the impact of these eigenvalues of $A$ on $h^*$ is mild and that the eigenvalue-dependent constant terms in Corollary 3 in our vector analysis do not significantly affect the optimal step-size $h^*$. The eigenvalues do influence the values of the MSE achieved in each system. MSE decreases as the magnitude of the eigenvalue increases. In the infinite horizon system, the horizon needs to be large enough to manage truncation error while simultaneously being small such that we can run multiple rollouts. We choose $\gamma$ large enough such that we can learn a good estimate of $V$. We set $T = 1/(1 - \gamma)$, which is commonly referred to as the effective horizon in the RL literature.

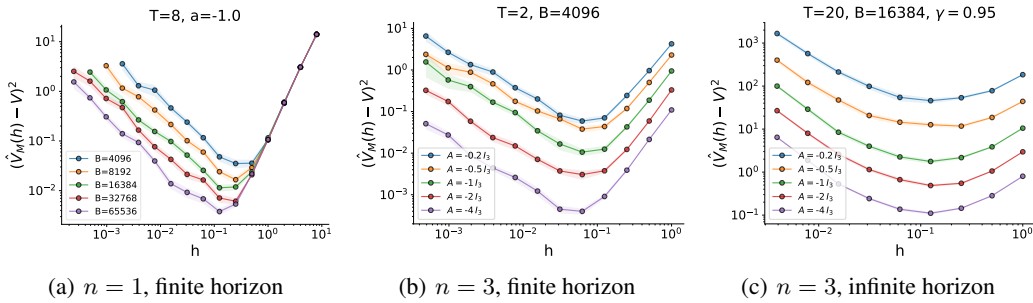

(a) $n = 1$, finite horizon      (b) $n = 3$, finite horizon      (c) $n = 3$, infinite horizon

Figure 1: Mean-squared error trade-off in linear quadratic systems of different dimension $n$. The left-most plot shows the dependence of the optimal step-size on the data budget; as expected the optimal step-size decreases with more data. Middle and right plot show the MSE for different drift matrices $A$. Note that the optimal step-size $h$ exhibits only a mild dependence on the scale of $A$.

## 4.2 NONLINEAR SYSTEMS

We empirically show that the trade-off identified in linear quadratic systems carries over to nonlinear systems, with more complex cost functions. We demonstrate it in several simulated nonlinear systems from OpenAI Gym (Brockman et al., 2016), including Pendulum, BipedalWalker and six MuJoCo (Todorov et al., 2012) environments: InvertedDoublePendulum, Pusher, Swimmer, Hopper, HalfCheetah and Ant. We note that the original environments all have a fixed temporal discretization $\delta t$, pre-chosen by the designer. To measure the effect of $h$, we first modify all environments to run at a small discretization $\delta t = 0.001$ as the proxy to the underlying continuous-time systems. We train a nonlinear policy parameterized by a neural network for each system, by the algorithm DAU (Tallec et al., 2019). This policy is used to gather episode data from the continuous-time system proxy at intervals of $\delta t = 0.001$ which are then down-sampled for different $h$ based on the ratio of $h$ and $\delta t$. This allowed to handle uniformly all environments, thus yielding the behaviour of the MSE with respect to the sampling time $h$ in the same interval. The policy is stable in the sense that it produces reasonable behavior (e.g., pendulum stays mostly upright; Ant walking forward etc) and not cause early termination of episodes (e.g., BipedalWalker does not fall), in the continuous-time system proxy. The results of the MSE of Monte-Carlo value estimation are shown in Fig. 2. Similar to the linear systems case, we vary the data budget $B$ and see how the MSE changes with the discretization $h$. We slightly abuse notations by using $V, \hat{V}$ to refer to the true and estimated sum of rewards instead of the cost. The true value of $V$ is approximated by averaging the sum of rewards observed at $\delta t = 0.001$ from $150k$ episodes. These environments fall under the finite horizon undiscounted setting. The system (and estimation) horizon $T$ of our experiments is chosen to be the physical time of $1k$ steps under the default $\delta t$ in the original environments (200 steps for Pendulum and 500 steps for BipedalWalker). Please refer to Appendix F for more details on the setup including $B, T, \delta t, h$.

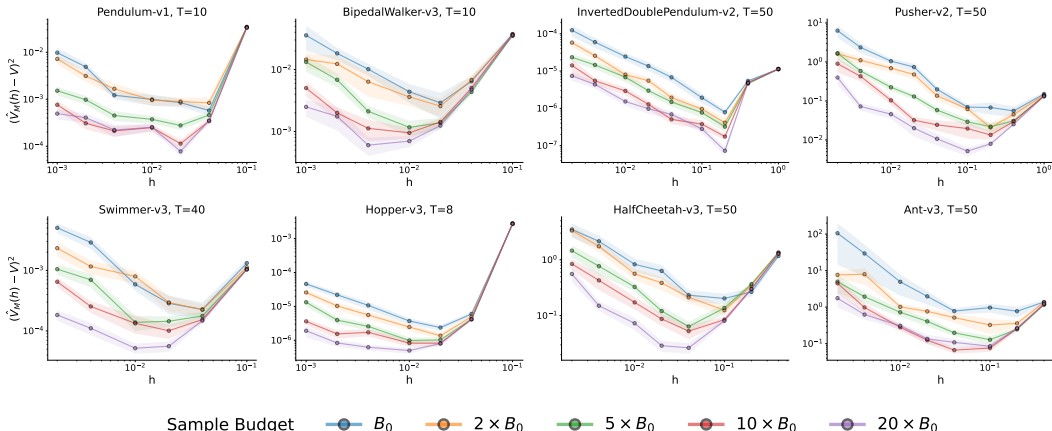

Figure 2: MSE of Monte-Carlo value estimation in nonlinear systems. The line and shaded region denote the sample mean and its standard error of $(\hat{V}_M(h) - V)^2$, from 30 random runs. $T$ is the horizon in physical time (seconds). $B_0$ denotes the environment-dependent base sample budget, chosen such that it gives a full episode for the smallest $h$ (see Appendix F). In almost all environments the optimal step-size depends on the data budget (with 'InvertedDoublePendulum-v2' being the only exception). In particular, the MSE as a function of $h$ shows a clear minimum for choosing the optimal step-size, which generally decreases as the data budget increases.

These system are stochastic in the starting state, while having deterministic dynamics. Despite the different settings from our analysis, a clear trade-off is evident in all systems.

**Optimal Step-Size in Nonlinear Systems** Fig. 3 plots the empirical $h^*$ over $B$ for all nonlinear environments, and fitted lines based on the relation $h^* = cTB^{-1/3}$ for finite horizon undiscounted linear systems described in Corollary 3. The plot shows that the analytical trade-off for linear systems is observed approximately even in the non-linear experiments. The constant $c$ depends on the system parameters (and the policy) and, as expected, varies with the environment. In our experiments, $c$ ranges from 0.02 to 0.2, which can serve as a starting point for optimizing the step-size in other experiments.

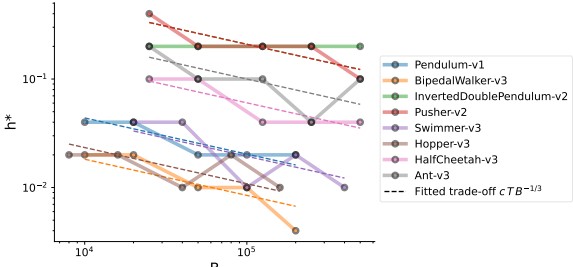

Figure 3: Empirical $h^*$ in nonlinear experiments (solid lines) aligns well with the analysis in Corollary 3 (dashed lines): $h^* = cTB^{-1/3}$. $c$ is environment dependent and estimated from the data by least squares.

## 5 CONCLUSION

We provide a precise characterization of the approximation, estimation and truncation errors incurred by Monte-Carlo value estimation in a Langevin dynamical system with quadratic cost. Our analysis reveals a fundamental bias-variance trade-off, modulated by the level of temporal discretization $h$. In a second step, we confirm in numerical simulations that the analysis accurately captures the trade-off in a precise, quantitative manner. In particular, we show that the trade-off carries over to non-linear environments such as the popular MuJoCo physics simulation. Our findings show that managing the temporal discretization level $h$ can greatly improve the quality of value estimation under a fixed data budget $B$. This has implication for practitioners, as in most environments that we encountered the step-size is typically pre-set and rarely changed. There are several interesting directions for future work, including considering policy optimization and other value estimation techniques such as temporal differencing and system identification. Another direction is to extend the analysis to non-linear systems via local linearization.

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

## A    WARM-UP: THE RIEMANN SUM APPROXIMATION

The Riemann sum approximation is a standard argument that we reproduce here for completeness. Let $g : [0, T] \to \mathbb{R}$ be a continuously differentiable function. Assume that we wish to approximate the integral $\int_0^T g(t)\ dt$ using the Riemann sum over $N = T/h$ elements, $\sum_{i=0}^{N-1} h\, g(ih)$. The difference is readily computed up to first order as follows:

$$
\begin{aligned}
D &= \int_0^T g(t)\ dt - \sum_{i=0}^{N-1} h\, g(ih) \\
&= \sum_{i=0}^{N-1} \int_{ih}^{(i+1)h} g(t) - g(ih)\, dt \\
&\leq \sum_{i=0}^{N-1} \int_0^h g'(ih)\, t + \mathcal{O}(t^2)\, dt \\
&= \frac{1}{2} \sum_{i=0}^{N-1} \left( g'(ih)h^2 + \mathcal{O}(h^3) \right)
\end{aligned}
$$

A naive bound is obtained as $D \leq \frac{1}{2} N \, h^2 \, \|g'\|_\infty + \mathcal{O}(Nh^3)$. Translated to a squared error, this explains the dependency $\mathcal{O}(N^2 h^4) = \mathcal{O}(T^2 h^2)$.

In the case of discounting, let $g(t) = \gamma^t f(t)$ and $g'(t) = \gamma^t(f(t) + f'(t))$. Hence, the previous display leads to the bound

$$
\begin{aligned}
D &\leq \frac{1}{2} \sum_{i=0}^{N-1} \gamma^{ih} \left( h^2 \|f(t) + f'(t)\|_\infty + \mathcal{O}(h^3) \right) \\
&= \frac{h^2(1 - \gamma^{Nh}) \, \|f(t) + f'(t)\|_\infty}{2(1 - \gamma^h)} + \mathcal{O}(h^3) \, .
\end{aligned}
$$

Overall, the squared error is now $\mathcal{O}(h^4(1 - \gamma^T)^2/(1 - \gamma^h)^2) = \mathcal{O}(h^2/\log(1/\gamma))$. Note that this alone does not explain the improvement of the order from $h^2$ to $h^4$, which requires that also $f(t)$ is decaying fast enough.

## B  MOMENT CALCULATIONS

Recall that the solution of the SDE in Eq. (9), with $x(0) = 0$, takes the following form:

$$
x(t) = \sigma \int_0^t e^{a(t-s)} \, \mathrm{d}w(s). \tag{20}
$$

An integral part of finding the mean-squared error of the Monte-Carlo estimator is the computation of the moments $\mathbb{E}\left[x(t)^2\right], \mathbb{E}\left[x(t)^4\right]$ and $\mathbb{E}\left[x(s)^2 x(t)^2\right]$ when $s \leq t$.

**Lemma 1.** *Let $x(t)$ be the solution of Eq. (9). The second moment of the state variable is*

$$
\mathbb{E}[x^2(t)] = \frac{\sigma^2}{2a} \left( e^{2at} - 1 \right) \, . \tag{21}
$$

*For the forth moment, we get:*

$$
\mathbb{E}[x(t)^4] = \frac{3\sigma^4}{4a^2} \left( e^{2at} - 1 \right)^2 \tag{22}
$$

*Assuming that $s \leq t$, we further get:*

$$
\mathbb{E}[x^2(s)x^2(t)] = \frac{\sigma^4}{4a^2}(e^{2as} - 1)e^{2at} \left\{ (e^{-2as} - e^{-2at}) + 3(1 - e^{-2as}) \right\} \, . \tag{23}
$$

*Proof.*

**(1)**  We start with the second moment $\mathbb{E}\left[x(t)^2\right]$.

$$
\mathbb{E}\left[x(t)^2\right] = \sigma^2 e^{2at} \mathbb{E}\left[ \left( \int_0^t e^{-as} dw(s) \right)^2 \right] = \sigma^2 e^{2at} \int_0^t e^{-2as} ds = \frac{\sigma^2}{2a}(e^{2at} - 1)
$$

The calculation makes use of the Itô isometry, which can be stated as:

$$
\mathbb{E}\left[ \left( \int_0^t z(s) \, \mathrm{d}w(s) \right)^2 \right] = \mathbb{E}\left[ \int_0^t z(s)^2 \, \mathrm{d}s \right], \tag{24}
$$

for any stochastic process $z(\cdot)$ adapted to the filtration induced by the Wiener process $w(\cdot)$.

**(2)**  Next we compute $\mathbb{E}\left[x(t)^4\right]$ through Itô's integral. Define $y(t) := \int_0^t e^{-au} \, \mathrm{d}w(u)$, so that $\mathrm{d}y(t) = e^{-at} \, \mathrm{d}w(t)$. Thus,

$$
\begin{aligned}
\mathrm{d}f(y(t)) &= f'(y(t)) \, \mathrm{d}y(t) + \frac{1}{2} f''(y(t)) \, (\mathrm{d}y(t))^2 \\
&= f'(y(t)) \, e^{-at} \, \mathrm{d}w(t) + \frac{1}{2} f''(y(t)) \, e^{-2at} \, \mathrm{d}t,
\end{aligned}
$$

for any $f\left(\cdot\right)$. By choosing $f\left(y\right) = y^4$:

$$f'\left(y\right) = 4y^3 \quad \text{and} \quad f''\left(y\right) = 12y^2.$$

Therefore, by integration and taking the expectation:

$$\mathbb{E}\left[f\left(y\left(t\right)\right)\right] = \mathbb{E}\left[\int_0^t f'\left(y\left(u\right)\right)e^{-au}\,\mathrm{d}w\left(u\right)\right] + \frac{1}{2}\mathbb{E}\left[\int_0^t f''\left(y\left(u\right)\right)e^{-2au}\,\mathrm{d}u\right]$$

$$= \underbrace{\mathbb{E}\left[\int_0^t 4\left(\int_0^u e^{-av}\,\mathrm{d}w\left(v\right)\right)^3 e^{-au}\,\mathrm{d}w\left(u\right)\right]}_{=0} + \frac{1}{2}\mathbb{E}\left[\int_0^t 12\left(\int_0^u e^{-av}\,\mathrm{d}w\left(v\right)\right)^2 e^{-2au}\,\mathrm{d}u\right]$$

$$= 6\mathbb{E}\left[\int_0^t \left(\int_0^u e^{-av}e^{-au}\,\mathrm{d}w\left(v\right)\right)^2\,\mathrm{d}u\right]$$

$$= 6\int_0^t \mathbb{E}\left[\left(\int_0^u e^{-av}e^{-au}\,\mathrm{d}w\left(v\right)\right)^2\right]\,\mathrm{d}u \qquad \text{(Itô isometry)}$$

$$= 6\int_0^t \int_0^u e^{-2av}e^{-2au}\,\mathrm{d}v\,\mathrm{d}u$$

$$= \int_0^t e^{-2au}\frac{1}{2a}\left(1 - e^{-2au}\right)\,\mathrm{d}u$$

$$= \frac{3}{4a^2}\left(1 - e^{-2at}\right)^2$$

From Eq. (20) it holds $x\left(t\right) = \sigma e^{at}y\left(t\right)$ so that the second part of the lemma follows.

**(3)** Lastly, we compute $\mathbb{E}\left[x(s)^2 x(t)^2\right]$ for $s \leq t$.

$$\mathbb{E}\left[x^2\left(s\right)x^2\left(t\right)\right] = \sigma^4 e^{2a(s+t)}\mathbb{E}\left[\left(\int_0^s e^{-au}\,\mathrm{d}w\left(u\right)\right)^2\left(\int_0^t e^{-au}\,\mathrm{d}w\left(u\right)\right)^2\right]$$

$$= \sigma^4 e^{2a(s+t)}\mathbb{E}\left[\left(\int_0^s e^{-au}\,\mathrm{d}w\left(u\right)\right)^2\left(\int_0^s e^{-au}\,\mathrm{d}w\left(u\right) + \int_s^t e^{-au}\,\mathrm{d}w\left(u\right)\right)^2\right]$$

$$= \sigma^4 e^{2a(s+t)}\left\{\underbrace{\mathbb{E}\left[\left(\int_0^s e^{-au}\,\mathrm{d}w\left(u\right)\right)^4\right]}_{\text{(i)}} + \underbrace{\mathbb{E}\left[\left(\int_0^s e^{-au}\,\mathrm{d}w\left(u\right)\right)^2\right]\mathbb{E}\left[\left(\int_s^t e^{-au}\,\mathrm{d}w\left(u\right)\right)^2\right]}_{\text{(ii)}}\right\}$$

Note that we computed (i) before. For (ii) it holds:

$$\mathbb{E}\left[\left(\int_0^s e^{-au}\,\mathrm{d}w\left(u\right)\right)^2\right] = \int_0^s e^{-2au}\,\mathrm{d}w\left(u\right)$$

$$= \frac{1}{2a}(1 - e^{-2as})$$

and

$$\mathbb{E}\left[\left(\int_s^t e^{-au}\,\mathrm{d}w\left(u\right)\right)^2\right] = \int_s^t e^{-2au}\,\mathrm{d}w\left(u\right)$$

$$= \frac{1}{2a}(e^{-2as} - e^{-2at})$$

Therefore, assuming $s \leq t$, it holds that (it is necessary to use Itô integration in this case):

$$\mathbb{E}\left[x^2\left(s\right)x^2\left(t\right)\right] = \sigma^4 e^{2a(s+t)}\left\{\frac{1}{4a^2}\left(1 - e^{-2as}\right)\left(e^{-2as} - e^{-2at}\right) + \frac{3}{4a^2}\left(1 - e^{-2as}\right)^2\right\}$$

$$= \frac{\sigma^4}{4a^2}(e^{2at} - 1)e^{2as}\left\{(e^{-2at} - e^{-2as}) + 3(1 - e^{-2at})\right\}.$$

$\square$

## C  CALCULATIONS OF THE MEAN-SQUARED ERROR

### C.1  UNDISCOUNTED, FINITE-HORIZON: PROOF OF THEOREM 1

*Proof.* We first note that

$$\mathbb{E}[\hat{V}_M(h)] = \frac{h}{M} \sum_{i=1}^{M} \sum_{k=0}^{N-1} \mathbb{E}[x_i^2(kh)] = h \sum_{k=0}^{N-1} \mathbb{E}[x^2(kh)]$$

where we denote $x(t) = x_1(t)$ for simplicity. Next we expand the mean-squared error

$$\begin{aligned}
\mathbb{E}[(\hat{V}_M(h) - V_T)^2] &= \mathbb{E}[\hat{V}_M^2(h)] - 2V_T \mathbb{E}[\hat{V}_M(h)] + V_T^2 \\
&= \frac{h^2}{M^2} \mathbb{E}\left[ \left( \sum_{i=1}^{M} \sum_{k=0}^{N-1} x_i^2(kh) \right)^2 \right] - 2V_T \mathbb{E}[\hat{V}_M(h)] + V_T^2 \\
&= \frac{h^2}{M^2} \sum_{i,j=1}^{M} \sum_{k,l=0}^{N-1} \mathbb{E}[x_i^2(kh) x_j^2(lh)] - 2V_T \mathbb{E}[\hat{V}_M(h)] + V_T^2 \\
&= \frac{h^2}{M} \sum_{k,l=0}^{N-1} \mathbb{E}[x^2(kh)x^2(lh)] + \frac{M^2 - M}{M^2} \mathbb{E}[\hat{V}_M(h)]^2 - 2V_T \mathbb{E}[\hat{V}_M(h)] + V_T^2
\end{aligned}$$

For the last equality, note that $\mathbb{E}[\hat{V}_M(h)]^2 = h^2 \sum_{k,l=0}^{N-1} \mathbb{E}[x^2(kh)]\mathbb{E}[x^2(lh)]$. It remains to compute the expressions. By Lemma 1 we have for the second moment of the state variable:

$$\mathbb{E}[x^2(t)] = \frac{\sigma^2}{2a} \left( e^{2at} - 1 \right) . \tag{25}$$

Assuming that $s \le t$, from the same lemma we get the following for the forth moments:

$$\mathbb{E}[x^2(s)x^2(t)] = \frac{\sigma^4}{4a^2} (e^{2as} - 1) e^{2at} \left\{ (e^{-2as} - e^{-2at}) + 3(1 - e^{-2as}) \right\} . \tag{26}$$

Note that by symmetry, a similar expression follows for $s \ge t$.

Using these expressions, for the expected cost we get

$$V_T = \int_0^T \mathbb{E}[x^2(t)]dt = \frac{\sigma^2}{2a} \int_0^T \left( e^{2at} - 1 \right) dt = \frac{\sigma^2}{2a} \left( \frac{e^{2aT} - 1}{2a} - T \right) \tag{27}$$

We remark that a similar expression was previously obtained in (Bijl et al., 2016, Theorem 3). Next, the expected estimated cost is

$$\mathbb{E}[\hat{V}_M(h)] = h \sum_{k=0}^{N-1} \mathbb{E}[x^2(kh)] = \frac{\sigma^2 h}{2a} \sum_{k=0}^{N-1} \left( e^{2akh} - 1 \right) = \frac{\sigma^2 h}{2a} \left[ \frac{1 - e^{2aT}}{1 - e^{2ah}} - N \right]$$

Lastly, it remains to compute the sum

$$\frac{h^2}{M} \sum_{k,l=0}^{N-1} \mathbb{E}[x^2(kh)x^2(lh)] = \frac{2h^2}{M} \sum_{k<l}^{N-1} \mathbb{E}[x^2(kh)x^2(lh)] + \frac{h^2}{M} \sum_{k=0}^{N-1} \mathbb{E}[x^4(kh)]$$

$$= \frac{\sigma^4 T \left( h^2 \left( e^{2aT} - 1 \right) \left( 8e^{2ah} + 3e^{2aT} + 1 \right) + T^2 \left( e^{2ah} - 1 \right)^2 - 2hT \left( e^{2ah} - 1 \right) \left( e^{2ah} + 5e^{2aT} \right) \right)}{4a^2 Bh \left( e^{2ah} - 1 \right)^2}$$

The last equality is a cumbersome calculation that involves nested geometric sums. We verified the result using symbolic computation. For reference we provide the notebooks containing all calculations in the supplementary material. It remains to collect all terms to get the final result.  □

## C.2 Undiscounted, Finite-Horizon: Step Size

Although the exact optimal step size $h^*$ can be obtained from Theorem 1 in practice, such exact $h^*$ doesn't have an explicit solution through Theorem 1. A trivial way to see the order of $h^*$ in terms of $B, a, T$ is finding the dominated term by using Taylor's expansion for exponential parts (which is true for any $h$) in Theorem 1. A proof of Corollary 2 is given as follows.

*Proof.* From Theorem 1, we compute the leading terms in $h$ of the least-squares error:

$$E_1(h, T, a) = \frac{\sigma^4(e^{2aT} - 1)^2}{16a^2} h^2 + \mathcal{O}(h^3), \tag{28}$$

$$\frac{E_2(h, T, a)}{B} = -\frac{\sigma^4 \left(4aT - e^{4aT} + e^{2aT}(8aT - 4) + 5\right)}{8a^4} \cdot \frac{1}{hB} + \frac{\sigma^4 T(1 - e^{4aT} + 4aTe^{2aT})}{4a^3 B}$$
$$- \frac{\sigma^4 Th \left(1 + 4aT + e^{2aT}(8aT + 4) - 5e^{4aT}\right)}{24a^2 B} - \frac{\sigma^4 Th^2(e^{4aT} - 1)}{12aB} + \mathcal{O}\left(h^3/B\right). \tag{29}$$

It is trivial to see, when $h \geq 1$, both Eq. (28) and Eq. (29) will blow up, and increase exponentially in $h$. Thus, a small $h < 1$ is considered to minimize $E_1(h, T, a) + \frac{E_2(h, T, a)}{B}$. Keeping the first term in both Eq. (28) and Eq. (29) and solving for the optimal $h^*$ yields the result. $\square$

A more precise approximation of $h^*$ than Corollary 2 is a minimizer of $E_1(h, T, a) + \frac{E_2(h, T, a)}{B}$ truncated at $\mathcal{O}(h^3)$:

$$h^*(a, T, B) = \frac{D_1}{3D_3} + \left(\frac{D_1^3}{3^3 D_3^3} - \frac{3D_2}{2a^2 D_3} - \sqrt{\frac{9D_2^2}{4a^4 D_3^2} - \frac{D_1^3 D_2}{9a^2 D_3^4}}\right)^{\frac{1}{3}}$$
$$+ \left(\frac{D_1^3}{3^3 D_3^3} - \frac{3D_2}{2a^2 D_3} + \sqrt{\frac{9D_2^2}{4a^4 D_3^2} - \frac{D_1^3 D_2}{9a^2 D_3^4}}\right)^{\frac{1}{3}}, \tag{30}$$

where

$$D_1 = T \left(1 + 4aT + e^{2aT}(8aT + 4) - 5e^{4aT}\right),$$
$$D_2 = T \left(4aT - e^{4aT} + e^{2aT}(8aT - 4) + 5\right),$$
$$D_3 = 3B(e^{2aT} - 1)^2 - 4aT(e^{4aT} - 1).$$

We can further express Eq. (30) in terms of $B$, as

$$h^*(B) = \left(-\frac{T \left(4aT - e^{4aT} + e^{2aT}(8aT - 4) + 5\right)}{a^2(e^{2aT} - 1)^2}\right)^{1/3} B^{-1/3}$$
$$+ \frac{T \left(1 + 4aT + e^{2aT}(8aT + 4) - 5e^{4aT}\right)}{9(e^{2aT} - 1)^2 B} + \frac{4aT(e^{2aT} + 1)D_2^{1/3}}{9a^{2/3}(e^{2aT} - 1)^{5/3}} B^{-4/3}$$
$$+ \frac{4aT^2(e^{2aT} + 1)D_1}{27(e^{2aT} - 1)^3} B^{-2} + \mathcal{O}(B^{-7/3}).$$

where the first term is exactly the result in Corollary 2.

## C.3 Finite-Horizon, Discounted

As stated in Section 3.1, adding discounting in the finite-horizon setting makes the mean-squared error more involved. In the regime where $h$ is small and $B$ is large, a Taylor expansion characterizes the error surface as follows:

$$\text{MSE}_T(h, B, \gamma) \approx \frac{\sigma^4 T}{\log(\gamma)(a + \log(\gamma))(2a + \log(\gamma))^2} \cdot \frac{1}{hB} + \frac{\sigma^4 \gamma^{2T}(e^{2aT} - 1)^2}{16a^2} \cdot h^2$$
$$+ \frac{\gamma^T \left(e^{2aT} - 1\right) \left(\gamma^T \left(e^{2aT}(2a + \log(\gamma)) - \log(\gamma)\right) - 2a\right)}{48a^2} \cdot h^3 + \frac{\sigma^4}{144} \cdot h^4 \tag{31}$$

The approximation shows only the lowest order terms for $1/(hB)$, $\gamma^T$ and $h$. The derivation is given in Lemma 2 below. The results shows that main trade-off between $h$ and $B$ persists also for the discounted objective, as long as $\gamma^T$ is treated as a constant relative to $h^2$ and $1/hB$. In the limit where $\gamma^T$ becomes small (e.g. $\gamma^T = o(h^4)$) the nature of the trade-off changes in that the approximation error improves to $\mathcal{O}(h^4)$. This can be understood from the fact that under geometric discounting combined with a decaying process, the sum of $N = T/h$ estimation errors do not suffer a factor $N$, thereby removing a factor of $1/h$ from the (non-squared) approximation error (see Appendix A for a more detailed explanation).

**Lemma 2** (Finite-horizon, discounted). *In the finite-horizon with a discount factor $\gamma \in (0,1]$ setting, the mean-squared error of the Monte-Carlo estimator is*

$$\text{MSE}_T(h, B, \gamma) = E_1(h, T, a, \gamma) + \frac{E_2(h, T, a, \gamma)}{B},$$

*where*

$$E_1(h, T, a, \gamma) = C_1(T, \gamma, a)\sigma^4 h^2 + C_2(T, \gamma, a)\sigma^4 h^3 + \left(\frac{1}{144} + C_3(T, \gamma, a)\right)\sigma^4 h^4 + \mathcal{O}(h^5),$$

$$E_2(h, T, a, \gamma) = \frac{\sigma^4 T + \gamma^T C_4(T, \gamma, a)}{\log(\gamma)(a + \log(\gamma))(2a + \log(\gamma))^2 h} + \gamma^T \mathcal{O}(1),$$

$$C_1(T, \gamma, a) = \frac{\gamma^{2T}\left(e^{2aT} - 1\right)^2}{16a^2},$$

$$C_2(T, \gamma, a) = \frac{\gamma^T\left(e^{2aT} - 1\right)\left(\gamma^T\left(e^{2aT}(2a + \log(\gamma)) - \log(\gamma)\right) - 2a\right)}{48a^2},$$

$$C_3(T, \gamma, a) = \frac{\sigma^4 \gamma^T\left[\gamma^T\left(e^{2aT}(2a + \log(\gamma)) - \log(\gamma)\right)^2 - 4a\left(e^{2aT}(2a + \log(\gamma)) - \log(\gamma)\right)\right]}{576a^2},$$

$C_4(T, \gamma, a)$ *is some finite constant of $(T, \gamma, a)$ that includes $\gamma^T$ as a factor.*

*Proof.* The proof follows the similar computations as those in the previous proof with a new expected cost as follows. In particular, using Lemma 1, we get

$$V_T = \int_0^T \gamma^t \mathbb{E}[x^2(t)]dt = \frac{\sigma^2}{2a}\left(\frac{\gamma^T e^{2aT} - 1}{\log(\gamma) + 2a} - \frac{\gamma^T - 1}{\log(\gamma)}\right) \tag{32}$$

Furthermore, the expected estimated cost is

$$\mathbb{E}[\hat{V}_M(h)] = \frac{\sigma^2 h}{2a}\sum_{k=0}^{N-1}\gamma^{kh}\left(e^{2akh} - 1\right) = \frac{\sigma^2 h}{2a}\left(\frac{1 - \gamma^T e^{2aT}}{1 - \gamma^h e^{2ah}} - \frac{1 - \gamma^T}{1 - \gamma^h}\right).$$

Finally, the sum containing the forth order cross-moments is

$$\frac{h^2}{M}\sum_{k,l=0}^{N-1}\gamma^{kh+lh}\mathbb{E}[x^2(kh)x^2(lh)] = \frac{2h^2}{M}\sum_{k<l}^{N-1}\gamma^{kh+lh}\mathbb{E}[x^2(kh)x^2(lh)] + \frac{h^2}{M}\sum_{k=0}^{N-1}\gamma^{2kh}\mathbb{E}[x^4(kh)].$$

While not impossible to calculate on paper, a written derivation is beyond the scope of this work. Instead, we rely on symbolic computation to obtain the expression and corresponding Taylor approximations. The notebooks containing all derivations are provided in the supplementary material. $\qquad\square$

## C.4 INFINITE HORIZON: PROOF OF THEOREM 3

*Proof.* The proof relies on the decomposition provided in Eq. (17). It only remains to compute the following cross term.

$$\mathbb{E}\left[\hat{V}_M(h) - V_T\right] V_{T,\infty}$$

$$= \frac{\sigma^4}{2a}\left(\frac{\gamma^T}{\log(\gamma)} - \frac{\gamma^T e^{2aT}}{\log(\gamma) + 2a}\right)\left[\frac{h}{2a}\left(\frac{1 - \gamma^T e^{2aT}}{1 - \gamma^h e^{2ah}} - \frac{1 - \gamma^T}{1 - \gamma^h}\right) - \frac{1}{2a}\left(\frac{\gamma^T e^{2aT} - 1}{\log(\gamma) + 2a} - \frac{\gamma^T - 1}{\log(\gamma)}\right)\right]$$

$$= \frac{\sigma^4 \gamma^{2T}\left(e^{2aT} - 1\right)\left(\log(\gamma)\left(e^{2aT} - 1\right) - 2a\right)}{8a^2 \log(\gamma)\left(2a + \log(\gamma)\right)}h+$$

$$\frac{\sigma^4 \gamma^T\left(2a + \log(\gamma) - e^{2aT}\log(\gamma)\right)\left(2a\left(\gamma^T e^{2aT} - 1\right) + \gamma^T \log(\gamma)\left(e^{2aT} - 1\right)\right)}{48a^2 \log(\gamma)\left(2a + \log(\gamma)\right)}h^2 + \mathcal{O}(h^3)\,.$$

Thus, the mean-squared error $\text{MSE}_\infty(h, B, T, \gamma) = \mathbb{E}\left[(\hat{V}_M(h) - V_\infty)^2\right]$ is obtained by combining the above computation with Eq. (16) and Lemma 2. $\qquad\square$

# D VECTOR CASE ANALYSIS

## D.1 FINITE-HORIZON, UNDISCOUNTED: PROOF OF THEOREM 2

*Proof.* Consider the n-dimensional system that the solution of the trajectory of $X(t)$ is

$$X(t) = \sigma \int_0^t e^{A(t-s)}\,\mathrm{d}W(t)\,.$$

Since $A$ is a diagonalizable matrix, we can decompose $A$ as $A = P^{-1}DP$, where $P$ is a invertible matrix (not necessarily to be orthogonal) and $D$ is a diagonal matrix whose diagonal entries $(\lambda_1, \cdots, \lambda_n)$ are corresponding to the eigenvalues of the matrix $A$. Followed by which, we can decompose the matrix exponential of $A$ as:

$$e^{At} = P^{-1}e^{Dt}P\,.$$

Define the "diagonalized" process $\tilde{X}(\cdot)$ as:

$$PX(t) = P\sigma \int_0^t e^{A(t-s)}\,\mathrm{d}W(s)$$

$$= \sigma PP^{-1}\int_0^t e^{D(t-s)}P\,\mathrm{d}W(s)$$

$$= \sigma \int_0^t e^{D(t-s)}\,\mathrm{d}\tilde{W}(s) =: \tilde{X}(t)$$

where $\tilde{W}(s)$ is a Wiener process (with dependent components when $P$ is not orthogonal). This implies that $X(\cdot) = P^{-1}\tilde{X}(\cdot)$.

To see $\tilde{X}_i(t)$ clearly, we denote $P = [p_{ij}]_{i,j=1}^n$, and $\tilde{X}_i(t) = (\phi_1^{(i)}(t), \cdots, \phi_n^{(i)}(t))^\top$, then $\phi_l^{(i)}(t) = \sum_{j=1}^n p_{lj}\sigma \int_0^t e^{\lambda_l(t-s)}\,\mathrm{d}w_j^{(i)}(s)$ for each $l \in \{1, \cdots, n\}$. Particularly, in such an expression, $w_j^{(i)}(s)$ are independent Wiener processes for different $i$ or $j$. Correspondingly, $\tilde{X}(t) = (\phi_1(t), \cdots, \phi_n(t))^\top$, and $\phi_l(t) = \sum_{j=1}^n p_{lj}\sigma \int_0^t e^{\lambda_l(t-s)}\,\mathrm{d}w_j(s)$ for each $l \in \{1, \cdots, n\}$, where $w_j(s)$ are independent Wiener processes for different $j$.

By trace operation, we can rewrite $\hat{V}_M(h)$ as follows:

$$\hat{V}_M(h) = \frac{1}{M}\sum_{i=1}^M\sum_{k=0}^{N-1} hX(t_k)^\top QX(t_k)$$

$$= \text{tr}\left\{\frac{1}{M}\sum_{i=1}^M\sum_{k=0}^{N-1} h\tilde{X}(t_k)^\top P^{-\top}QP^{-1}\tilde{X}(t_k)\right\}$$

$$= \text{tr}\left\{P^{-\top}QP^{-1}\hat{\mathcal{V}}_M(h)\right\}\,,$$

where $\hat{\mathcal{V}}_M(h) = \frac{1}{M} \sum_{i=1}^{M} \sum_{k=0}^{N-1} h \tilde{X}(t_k) \tilde{X}(t_k)^\top \in \mathbb{R}^{n \times n}$.

Similarly, $V_T = \text{tr}\{P^{-\top} Q P^{-1} \mathcal{V}_T\}$, where $\mathcal{V}_T = \int_0^T \mathbb{E}[\tilde{X}(t)\tilde{X}(t)^\top]\,\mathrm{d}t$.

Therefore, the $\text{MSE}_T(h, B)$ can be written as

$$\text{MSE}_T(h, B) = \mathbb{E}\left[\left(\hat{V}_M(h) - V_T\right)^2\right] = \mathbb{E}\left[\text{tr}\left\{P^{-\top} Q P^{-1}\left(\hat{\mathcal{V}}_M(h) - \mathcal{V}_T\right)\right\}^2\right]. \tag{33}$$

For notional simplicity, we denote matrix $P^{-\top} Q P^{-1} =: B = [b_{lj}]_{l,j=1}^n$ and $\hat{\mathcal{V}}_M(h) - \mathcal{V}_T =: C = [c_{lj}]_{l,j=1}^n$.

Noting the fact that

$$\text{MSE}_T(h, B) = \mathbb{E}\left[\left(\sum_{l,j} b_{jl} c_{lj}\right)^2\right] = \sum_{l_1, j_1, l_2, j_2} b_{j_1 l_1} b_{j_2 l_2} \mathbb{E}\left[c_{l_1 j_1} c_{l_2 j_2}\right], \tag{34}$$

it is sufficient to find $\text{MSE}_T$ by only computing $\mathbb{E}\left[c_{l_1 j_1} c_{i_2 j_2}\right]$.

We first introduce the following expectations that are used in the computations. For any $s \leq t$

$$\mathbb{E}\left[\int_0^t e^{\lambda_1(t-u)}\,\mathrm{d}w(u) \int_0^s e^{\lambda_2(s-u)}\,\mathrm{d}w(u)\right] = \frac{e^{\lambda_1 t + \lambda_2 s}}{\lambda_1 + \lambda_2}\left(1 - e^{-(\lambda_1 + \lambda_2)s}\right), \tag{35}$$

$$\mathbb{E}\left[\int_0^s e^{\lambda_1(s-u)}\,\mathrm{d}w(u) \int_0^s e^{\lambda_2(s-u)}\,\mathrm{d}w(u) \int_0^t e^{\lambda_3(t-u)}\,\mathrm{d}w(u) \int_0^t e^{\lambda_4(t-u)}\,\mathrm{d}w(u)\right]$$

$$= e^{(\lambda_1 + \lambda_2)s + (\lambda_3 + \lambda_4)t}\left[\frac{1}{(\lambda_1 + \lambda_2)(\lambda_3 + \lambda_4)}\left(1 - e^{-(\lambda_1 + \lambda_2)s}\right)\left(1 - e^{-(\lambda_3 + \lambda_4)s}\right)\right.$$

$$+ \frac{1}{(\lambda_1 + \lambda_3)(\lambda_2 + \lambda_4)}\left(1 - e^{-(\lambda_1 + \lambda_3)s}\right)\left(1 - e^{-(\lambda_2 + \lambda_4)s}\right)$$

$$+ \frac{1}{(\lambda_1 + \lambda_4)(\lambda_2 + \lambda_3)}\left(1 - e^{-(\lambda_1 + \lambda_4)s}\right)\left(1 - e^{-(\lambda_2 + \lambda_3)s}\right)$$

$$\left.+ \frac{1}{(\lambda_1 + \lambda_2)(\lambda_3 + \lambda_4)}\left(1 - e^{-(\lambda_1 + \lambda_2)s}\right)\left(e^{-(\lambda_3 + \lambda_4)s} - e^{-(\lambda_3 + \lambda_4)t}\right)\right] \tag{36}$$

$$\int_0^T \mathbb{E}\left[\int_0^t e^{\lambda_1(t-u)}\,\mathrm{d}w(u) \int_0^s e^{\lambda_2(-u)}\,\mathrm{d}w(u)\right]\,\mathrm{d}t = \frac{e^{(\lambda_1 + \lambda_2)T} - 1 - (\lambda_1 + \lambda_2)T}{(\lambda_1 + \lambda_2)^2}. \tag{37}$$

By using the definitions of $\hat{\mathcal{V}}_M(h)$ and $\mathcal{V}_T$, it is trivial to see for any $l, j \in \{1, \cdots, n\}$

$$c_{lj} = \frac{1}{M} \sum_{i=1}^{M} \sum_{k=0}^{N-1} h\phi_l^{(i)}(kh)\phi_j^{(i)}(kh) - \int_0^T \mathbb{E}[\phi_l(t)\phi_j(t)]\,\mathrm{d}t$$

$$= \frac{h\sigma^2}{M} \sum_{i=1}^{M} \sum_{k=0}^{N-1}\left(\sum_{\alpha=1}^n p_{l\alpha} \int_0^{kh} e^{\lambda_l(kh-s)}\,\mathrm{d}w_\alpha^{(i)}(s)\right)\left(\sum_{\alpha=1}^n p_{j\alpha} \int_0^{kh} e^{\lambda_j(kh-s)}\,\mathrm{d}w_\alpha^{(i)}(s)\right)$$

$$- \sigma^2 \int_0^T \mathbb{E}\left[\left(\sum_{\alpha=1}^n p_{l\alpha} \int_0^t e^{\lambda_l(t-s)}\,\mathrm{d}w_\alpha(s)\right)\left(\sum_{\alpha=1}^n p_{j\alpha} \int_0^t e^{\lambda_j(t-s)}\,\mathrm{d}w_\alpha(s)\right)\right]\,\mathrm{d}t$$

$$= \sum_{\alpha=1}^n p_{l\alpha} p_{j\alpha}\left[\frac{h\sigma^2}{M} \sum_{i=1}^{M} \sum_{k=0}^{N-1}\left(\int_0^{kh} e^{\lambda_l(kh-s)}\,\mathrm{d}w_\alpha^{(i)}(s)\right)\left(\int_0^{kh} e^{\lambda_j(kh-s)}\,\mathrm{d}w_\alpha^{(i)}(s)\right)\right.$$

$$\left.- \sigma^2 \int_0^T \mathbb{E}\left[\left(\int_0^t e^{\lambda_l(t-s)}\,\mathrm{d}w_\alpha(s)\right)\left(\int_0^t e^{\lambda_j(t-s)}\,\mathrm{d}w_\alpha(s)\right)\right]\,\mathrm{d}t\right] +$$

$$\sum_{\alpha \neq \beta} p_{l\alpha} p_{j\beta}\left[\frac{h\sigma^2}{M} \sum_{i=1}^{M} \sum_{k=0}^{N-1}\left(\int_0^{kh} e^{\lambda_l(kh-s)}\,\mathrm{d}w_\alpha^{(i)}(s)\right)\left(\int_0^{kh} e^{\lambda_j(kh-s)}\,\mathrm{d}w_\beta^{(i)}(s)\right)\right],$$

where the last equation is due to the fact that for $\alpha \neq \beta$

$$\mathbb{E}\left[\left(\int_0^t e^{\lambda_l(t-s)}\,\mathrm{d}w_\alpha(s)\right)\left(\int_0^t e^{\lambda_j(t-s)}\,\mathrm{d}w_\beta(s)\right)\right] = 0\,.$$

Thus, for any $l_1, l_2, j_1, j_2 \in \{1, \cdots, n\}$,

$$\begin{aligned}
\mathbb{E}\left[c_{l_1 j_1} c_{l_2, j_2}\right] &= \sum_{\alpha=1}^n p_{l_1\alpha} p_{j_1\alpha} p_{l_2\alpha} p_{j_2\alpha} \sigma^4 \mathcal{I}_1\left(M, h, T, \lambda_{l_1}, \lambda_{j_1}, \lambda_{l_2}, \lambda_{j_2}, \alpha\right) \\
&+ \sum_{\alpha\neq\beta}^n p_{l_1\alpha} p_{j_1\alpha} p_{l_2\beta} p_{j_2\beta} \sigma^4 \mathcal{I}_2\left(M, h, T, \lambda_{l_1}, \lambda_{j_1}, \lambda_{l_2}, \lambda_{j_2}, \alpha, \beta\right) \\
&+ \sum_{\alpha\neq\beta}^n p_{l_1\alpha} p_{j_1\beta} p_{l_2\alpha} p_{j_2\beta} \sigma^4 \mathcal{I}_3\left(M, h, T, \lambda_{l_1}, \lambda_{j_1}, \lambda_{l_2}, \lambda_{j_2}, \alpha, \beta\right),
\end{aligned} \tag{38}$$

where

$$\begin{aligned}
&\mathcal{I}_1\left(M, h, T, \lambda_{l_1}, \lambda_{j_1}, \lambda_{l_2}, \lambda_{j_2}, \alpha\right) \\
&= \mathbb{E}\Bigg\{\Bigg[\frac{h}{M}\sum_{i=1}^M\sum_{k=0}^{N-1}\left(\int_0^{kh} e^{\lambda_{l_1}(kh-s)}\,\mathrm{d}w_\alpha^{(i)}(s)\right)\left(\int_0^{kh} e^{\lambda_{j_1}(kh-s)}\,\mathrm{d}w_\alpha^{(i)}(s)\right) \\
&\quad - \int_0^T \mathbb{E}\left[\left(\int_0^t e^{\lambda_{l_1}(t-s)}\,\mathrm{d}w_\alpha(s)\right)\left(\int_0^t e^{\lambda_{j_1}(t-s)}\,\mathrm{d}w_\alpha(s)\right)\right]\,\mathrm{d}t\Bigg] \times \\
&\quad \Bigg[\frac{h}{M}\sum_{i=1}^M\sum_{k=0}^{N-1}\left(\int_0^{kh} e^{\lambda_{l_2}(kh-s)}\,\mathrm{d}w_\alpha^{(i)}(s)\right)\left(\int_0^{kh} e^{\lambda_{j_2}(kh-s)}\,\mathrm{d}w_\alpha^{(i)}(s)\right) \\
&\quad - \int_0^T \mathbb{E}\left[\left(\int_0^t e^{\lambda_{l_2}(t-s)}\,\mathrm{d}w_\alpha(s)\right)\left(\int_0^t e^{\lambda_{j_2}(t-s)}\,\mathrm{d}w_\alpha(s)\right)\right]\,\mathrm{d}t\Bigg]\Bigg\},
\end{aligned}$$

and

$$\begin{aligned}
&\mathcal{I}_2\left(M, h, T, \lambda_{l_1}, \lambda_{j_1}, \lambda_{l_2}, \lambda_{j_2}, \alpha, \beta\right) \\
&= \mathbb{E}\Bigg\{\Bigg[\frac{h}{M}\sum_{i=1}^M\sum_{k=0}^{N-1}\left(\int_0^{kh} e^{\lambda_{l_1}(kh-s)}\,\mathrm{d}w_\alpha^{(i)}(s)\right)\left(\int_0^{kh} e^{\lambda_{j_1}(kh-s)}\,\mathrm{d}w_\alpha^{(i)}(s)\right) \\
&\quad - \int_0^T \mathbb{E}\left[\left(\int_0^t e^{\lambda_{l_1}(t-s)}\,\mathrm{d}w_\alpha(s)\right)\left(\int_0^t e^{\lambda_{j_1}(t-s)}\,\mathrm{d}w_\alpha(s)\right)\right]\,\mathrm{d}t\Bigg] \times \\
&\quad \Bigg[\frac{h}{M}\sum_{i=1}^M\sum_{k=0}^{N-1}\left(\int_0^{kh} e^{\lambda_{l_2}(kh-s)}\,\mathrm{d}w_\beta^{(i)}(s)\right)\left(\int_0^{kh} e^{\lambda_{j_2}(kh-s)}\,\mathrm{d}w_\beta^{(i)}(s)\right) \\
&\quad - \int_0^T \mathbb{E}\left[\left(\int_0^t e^{\lambda_{l_2}(t-s)}\,\mathrm{d}w_\beta(s)\right)\left(\int_0^t e^{\lambda_{j_2}(t-s)}\,\mathrm{d}w_\beta(s)\right)\right]\,\mathrm{d}t\Bigg]\Bigg\}
\end{aligned}$$

and

$$\begin{aligned}
&\mathcal{I}_3\left(M, h, T, \lambda_{l_1}, \lambda_{j_1}, \lambda_{l_2}, \lambda_{j_2}, \alpha, \beta\right) \\
&= \mathbb{E}\Bigg\{\Bigg[\frac{h}{M}\sum_{i=1}^M\sum_{k=0}^{N-1}\left(\int_0^{kh} e^{\lambda_{l_1}(kh-s)}\,\mathrm{d}w_\alpha^{(i)}(s)\right)\left(\int_0^{kh} e^{\lambda_{j_1}(kh-s)}\,\mathrm{d}w_\beta^{(i)}(s)\right)\Bigg] \times \\
&\quad \Bigg[\frac{h}{M}\sum_{i=1}^M\sum_{k=0}^{N-1}\left(\int_0^{kh} e^{\lambda_{l_2}(kh-s)}\,\mathrm{d}w_\alpha^{(i)}(s)\right)\left(\int_0^{kh} e^{\lambda_{j_2}(kh-s)}\,\mathrm{d}w_\beta^{(i)}(s)\right)\Bigg]\Bigg\}.
\end{aligned}$$

Note that $w_\alpha^{(i)}$ and $w_\beta^{(i)}$ are independent for $\alpha \neq \beta$. By using the expectations Eqs. (35) and (37), we can further obtain $\mathcal{I}_2\left(M, h, T, \lambda_{l_1}, \lambda_{j_1}, \lambda_{l_2}, \lambda_{j_2}, \alpha, \beta\right)$ as

$$
\begin{aligned}
& \mathcal{I}_2\left(M, h, T, \lambda_{l_1}, \lambda_{j_1}, \lambda_{l_2}, \lambda_{j_2}, \alpha, \beta\right) \\
& = \left[\frac{h}{\left(\lambda_{l_1} + \lambda_{j_1}\right)}\left(\frac{1 - e^{\left(\lambda_{l_1} + \lambda_{j_1}\right)T}}{1 - e^{\left(\lambda_{l_1} + \lambda_{j_1}\right)h}} - \frac{T}{h}\right) - \frac{1}{\left(\lambda_{l_1} + \lambda_{j_1}\right)^2}\left(e^{\left(\lambda_{l_1} + \lambda_{j_1}\right)T} - 1 - \left(\lambda_{l_1} + \lambda_{j_1}\right)T\right)\right] \\
& \times \left[\frac{h}{\left(\lambda_{l_2} + \lambda_{j_2}\right)}\left(\frac{1 - e^{\left(\lambda_{l_2} + \lambda_{j_2}\right)T}}{1 - e^{\left(\lambda_{l_2} + \lambda_{j_2}\right)h}} - \frac{T}{h}\right) - \frac{1}{\left(\lambda_{l_2} + \lambda_{j_2}\right)^2}\left(e^{\left(\lambda_{l_2} + \lambda_{j_2}\right)T} - 1 - \left(\lambda_{l_2} + \lambda_{j_2}\right)T\right)\right].
\end{aligned}
$$

In the following computations, we will use $\bar{C}$ and $C(\lambda_{l_1}, \lambda_{j_1}, \lambda_{l_2}, \lambda_{j_2})$ to represent some constants that are not depending on $h, T, B$.

The expectation $\mathcal{I}_1\left(M, h, T, \lambda_{l_1}, \lambda_{j_1}, \lambda_{l_2}, \lambda_{j_2}, \alpha\right)$ is computed exactly the same way as in the proof of Theorem 1 by using the expectation results Eq. (35) and Eq. (36). Notice that the expectation result Eq. (35) (when $s = t$) has the same order in $t$ as the expectation Eq. (25). Moreover, the two expectations Eq. (36) and Eq. (26) have the same orders in $s$ and $t$. Thus, $\mathcal{I}_1\left(M, h, T, \lambda_{l_1}, \lambda_{j_1}, \lambda_{l_2}, \lambda_{j_2}, \alpha\right)$ has the same orders in $h, T, B$ as the scalar MSE, i.e.

$$
\begin{aligned}
\mathcal{I}_1\left(M, h, T, \lambda_{l_1}, \lambda_{j_1}, \lambda_{l_2}, \lambda_{j_2}, \alpha\right) = & \left(\bar{C}_1 + C_1\left(\lambda_{l_1}, \lambda_{j_1}, \lambda_{l_2}, \lambda_{j_2}\right)\mathcal{O}(T)\right)T^2 h^2 + \mathcal{O}(h^3) \\
& + \left(\bar{C}_2 + C_2\left(\lambda_{l_1}, \lambda_{j_1}, \lambda_{l_2}, \lambda_{j_2}\right)\mathcal{O}(T)\right)\frac{T^5}{hB} + \mathcal{O}\left(\frac{1}{B}\right)
\end{aligned}
$$

The expectation $\mathcal{I}_2\left(M, h, T, \lambda_{l_1}, \lambda_{j_1}, \lambda_{l_2}, \lambda_{j_2}, \alpha, \beta\right)$ can be computed directly and has the result:

$$
\begin{aligned}
\mathcal{I}_2\left(M, h, T, \lambda_{l_1}, \lambda_{j_1}, \lambda_{l_2}, \lambda_{j_2}, \alpha, \beta\right) & = \frac{\left(e^{\left(\lambda_{l_1} + \lambda_{j_1}\right)T} - 1\right)\left(e^{\left(\lambda_{l_2} + \lambda_{j_2}\right)T} - 1\right)h^2}{4\left(\lambda_{l_1} + \lambda_{j_1}\right)\left(\lambda_{l_2} + \lambda_{j_2}\right)} + \mathcal{O}(h^3) \\
& = \left(\frac{1}{4}T^2 + C_3(\lambda_{l_1}, \lambda_{j_1}, \lambda_{l_2}, \lambda_{j_2})\mathcal{O}(T^3)\right)h^2 + \mathcal{O}(h^3).
\end{aligned}
$$

The expectation $\mathcal{I}_3\left(M, h, T, \lambda_{l_1}, \lambda_{j_1}, \lambda_{l_2}, \lambda_{j_2}, \alpha, \beta\right)$ can be computed as follows:

$$
\begin{aligned}
& \mathcal{I}_3\left(M, h, T, \lambda_{l_1}, \lambda_{j_1}, \lambda_{l_2}, \lambda_{j_2}, \alpha, \beta\right) \\
& = \frac{h^2}{M}\sum_{k=0}^{n}\frac{\left(e^{\left(\lambda_{l_1} + \lambda_{l_2}\right)kh} - 1\right)\left(e^{\left(\lambda_{j_1} + \lambda_{j_2}\right)kh} - 1\right)h^2}{\left(\lambda_{l_1} + \lambda_{l_2}\right)\left(\lambda_{j_1} + \lambda_{j_2}\right)} + \\
& \frac{h^2}{M}\sum_{k<q}\frac{e^{\lambda_{l_1}kh + \lambda_{l_2}qh + \lambda_{j_1}kh + \lambda_{j_2}qh}}{\left(\lambda_{l_1} + \lambda_{l_2}\right)\left(\lambda_{j_1} + \lambda_{j_2}\right)}\left(1 - e^{-\left(\lambda_{l_1} + \lambda_{l_2}\right)kh}\right)\left(1 - e^{-\left(\lambda_{j_1} + \lambda_{j_2}\right)kh}\right) \\
& \frac{h^2}{M}\sum_{k<q}\frac{e^{\lambda_{l_1}qh + \lambda_{l_2}kh + \lambda_{j_1}qh + \lambda_{j_2}kh}}{\left(\lambda_{l_1} + \lambda_{l_2}\right)\left(\lambda_{j_1} + \lambda_{j_2}\right)}\left(1 - e^{-\left(\lambda_{l_1} + \lambda_{l_2}\right)kh}\right)\left(1 - e^{-\left(\lambda_{j_1} + \lambda_{j_2}\right)kh}\right) \\
& = \left(\bar{C}_4 + C_4\left(\lambda_{l_1}, \lambda_{j_1}, \lambda_{l_2}, \lambda_{j_2}\right)\mathcal{O}(T)\right)\frac{T^5}{hB} + \mathcal{O}\left(\frac{1}{B}\right).
\end{aligned}
$$

Thus, the final result is obtained by the expression of MSE in Eq. (34), Eq. (38) and the above computations. Again, we rely on symbolic computation to obtain the expression and corresponding Taylor approximations and include the notebooks of all derivations in the supplementary material.
$\square$

The extension from Theorem 2 to the discounted finite-horizon results can be done in the same way as in the above proof (add the discount factor $\gamma$ in $\hat{V}_M$) by using the expectation cost for any $\lambda_1$ and $\lambda_2$:

$$
\begin{aligned}
& \int_0^T \gamma^t \mathbb{E}\left[\int_0^t e^{\lambda_1(t-u)}\,\mathrm{d}w(u)\int_0^s e^{\lambda_2(-u)}\,\mathrm{d}w(u)\right]\,\mathrm{d}t \\
& = \frac{1}{\left(\lambda_1 + \lambda_2\right)}\left(\frac{\gamma^T e^{\left(\lambda_1 + \lambda_2\right)T} - 1}{\log(\gamma) + \left(\lambda_1 + \lambda_2\right)} - \frac{\gamma^T - 1}{\log(\gamma)}\right).
\end{aligned}
$$

### D.2 PROOF OF COROLLARY 4

*Proof.* We shall follow the similar proof as in the proof of Theorem 2 and the proof of Theorem 3. Continuing from Eq. (38), in infinite-horizon discounted setting, we have

$$
\begin{aligned}
&\mathcal{I}_1\left(M, h, T, \lambda_{l_1}, \lambda_{j_1}, \lambda_{l_2}, \lambda_{j_2}, \gamma, \alpha\right) \\
&= \mathbb{E}\Bigg\{\left[\frac{h}{M} \sum_{i=1}^{M} \sum_{k=0}^{N-1} \gamma^{kh}\left(\int_0^{kh} e^{\lambda_{l_1}(kh-s)}\, \mathrm{d}w_\alpha^{(i)}(s)\right)\left(\int_0^{kh} e^{\lambda_{j_1}(kh-s)}\, \mathrm{d}w_\alpha^{(i)}(s)\right)\right. \\
&\quad \left. -\int_0^\infty \gamma^t \mathbb{E}\left[\left(\int_0^t e^{\lambda_{l_1}(t-s)}\, \mathrm{d}w_\alpha(s)\right)\left(\int_0^t e^{\lambda_{j_1}(t-s)}\, \mathrm{d}w_\alpha(s)\right)\right]\, \mathrm{d}t\right] \times \\
&\quad \left[\frac{h}{M} \sum_{i=1}^{M} \sum_{k=0}^{N-1} \gamma^{kh}\left(\int_0^{kh} e^{\lambda_{l_2}(kh-s)}\, \mathrm{d}w_\alpha^{(i)}(s)\right)\left(\int_0^{kh} e^{\lambda_{j_2}(kh-s)}\, \mathrm{d}w_\alpha^{(i)}(s)\right)\right. \\
&\quad \left. -\int_0^\infty \gamma^t \mathbb{E}\left[\left(\int_0^t e^{\lambda_{l_2}(t-s)}\, \mathrm{d}w_\alpha(s)\right)\left(\int_0^t e^{\lambda_{j_2}(t-s)}\, \mathrm{d}w_\alpha(s)\right)\right]\, \mathrm{d}t\right]\Bigg\},
\end{aligned}
$$

and

$$
\begin{aligned}
&\mathcal{I}_2\left(M, h, T, \lambda_{l_1}, \lambda_{j_1}, \lambda_{l_2}, \lambda_{j_2}, \gamma, \alpha, \beta\right) \\
&= \left[\frac{h}{(\lambda_{l_1}+\lambda_{j_1})}\left(\frac{1-\gamma^T e^{(\lambda_{l_1}+\lambda_{j_1})T}}{1-\gamma^h e^{(\lambda_{l_1}+\lambda_{j_1})h}}-\frac{1-\gamma^T}{1-\gamma^h}\right)-\frac{1}{(\lambda_{l_1}+\lambda_{j_1})}\left(\frac{1}{\log(\gamma)}-\frac{1}{\log(\gamma)+\lambda_{l_1}+\lambda_{j_1}}\right)\right] \\
&\times \left[\frac{h}{(\lambda_{l_2}+\lambda_{j_2})}\left(\frac{1-\gamma^T e^{(\lambda_{l_2}+\lambda_{j_2})T}}{1-\gamma^h e^{(\lambda_{l_2}+\lambda_{j_2})h}}-\frac{1-\gamma^T}{1-\gamma^h}\right)-\frac{1}{(\lambda_{l_2}+\lambda_{j_2})}\left(\frac{1}{\log(\gamma)}-\frac{1}{\log(\gamma)+\lambda_{l_2}+\lambda_{j_2}}\right)\right],
\end{aligned}
$$

and

$$
\begin{aligned}
&\mathcal{I}_3\left(M, h, T, \lambda_{l_1}, \lambda_{j_1}, \lambda_{l_2}, \lambda_{j_2}, r, \alpha, \beta\right) \\
&= \mathbb{E}\Bigg\{\left[\frac{h}{M} \sum_{i=1}^{M} \sum_{k=0}^{N-1} \gamma^{kh}\left(\int_0^{kh} e^{\lambda_{l_1}(kh-s)}\, \mathrm{d}w_\alpha^{(i)}(s)\right)\left(\int_0^{kh} e^{\lambda_{j_1}(kh-s)}\, \mathrm{d}w_\beta^{(i)}(s)\right)\right] \times \\
&\quad \left[\frac{h}{M} \sum_{i=1}^{M} \sum_{k=0}^{N-1} \gamma^{kh}\left(\int_0^{kh} e^{\lambda_{l_2}(kh-s)}\, \mathrm{d}w_\alpha^{(i)}(s)\right)\left(\int_0^{kh} e^{\lambda_{j_2}(kh-s)}\, \mathrm{d}w_\beta^{(i)}(s)\right)\right]\Bigg\}.
\end{aligned}
$$

Similar arguments as in proof of Theorem 2, we can conclude $\mathcal{I}_1\left(M, h, T, \lambda_{l_1}, \lambda_{j_1}, \lambda_{l_2}, \lambda_{j_2}, \alpha\right)$ has the same orders in $h, B, T$ as the MSE result in Theorem 3.

Moreover, let $C_i(\lambda_{l_1}, \lambda_{j_1}, \lambda_{l_2}, \lambda_{j_2}, \gamma, T)$'s are some constants that depend on $\lambda_{l_1}, \lambda_{j_1}, \lambda_{l_2}, \lambda_{j_2}, \gamma, T$, then

$$
\begin{aligned}
&\mathcal{I}_2\left(M, h, T, \lambda_{l_1}, \lambda_{j_1}, \lambda_{l_2}, \lambda_{j_2}, \gamma, \alpha, \beta\right) \\
&\quad = \sigma^4 \gamma^{2T}\left(C_1(\lambda_{l_1}, \lambda_{j_1}, \lambda_{l_2}, \lambda_{j_2}, \gamma, T)+C_2(\lambda_{l_1}, \lambda_{j_1}, \lambda_{l_2}, \lambda_{j_2}, \gamma, T)h\right) \\
&\quad\quad +\sigma^4 \gamma^T\left(C_3(\lambda_{l_1}, \lambda_{j_1}, \lambda_{l_2}, \lambda_{j_2}, \gamma, T)h^2+C_4(\lambda_{l_1}, \lambda_{j_1}, \lambda_{l_2}, \lambda_{j_2}, \gamma, T)h^3\right) \\
&\quad\quad +\sigma^4\left(\frac{1}{144}+\gamma^T C_5(\lambda_{l_1}, \lambda_{j_1}, \lambda_{l_2}, \lambda_{j_2}, \gamma, T)\right)h^4+\mathcal{O}(h^5),
\end{aligned}
$$

and

$$\mathcal{I}_3\left(M, h, T, \lambda_{l_1}, \lambda_{j_1}, \lambda_{l_2}, \lambda_{j_2}, \gamma, \alpha, \beta\right)$$

$$= \frac{h^2}{M} \sum_{k=0}^{N-1} \frac{\left(e^{\left(\lambda_{l_1}+\lambda_{l_2}\right)kh}-1\right)\left(e^{\left(\lambda_{j_1}+\lambda_{j_2}\right)kh}-1\right)h^2\gamma^{2kh}}{\left(\lambda_{l_1}+\lambda_{l_2}\right)\left(\lambda_{j_1}+\lambda_{j_2}\right)}+$$

$$\frac{h^2}{M} \sum_{k<q} \frac{e^{\lambda_{l_1}kh+\lambda_{l_2}qh+\lambda_{j_1}kh+\lambda_{j_2}qh}}{\left(\lambda_{l_1}+\lambda_{l_2}\right)\left(\lambda_{j_1}+\lambda_{j_2}\right)}\left(1-e^{-\left(\lambda_{l_1}+\lambda_{l_2}\right)kh}\right)\left(1-e^{-\left(\lambda_{j_1}+\lambda_{j_2}\right)kh}\right)\gamma^{(k+q)h}$$

$$\frac{h^2}{M} \sum_{k<q} \frac{e^{\lambda_{l_1}qh+\lambda_{l_2}kh+\lambda_{j_1}qh+\lambda_{j_2}kh}}{\left(\lambda_{l_1}+\lambda_{l_2}\right)\left(\lambda_{j_1}+\lambda_{j_2}\right)}\left(1-e^{-\left(\lambda_{l_1}+\lambda_{l_2}\right)kh}\right)\left(1-e^{-\left(\lambda_{j_1}+\lambda_{j_2}\right)kh}\right)\gamma^{(k+q)h}$$

$$= C_6\left(\lambda_{l_1}, \lambda_{j_1}, \lambda_{l_2}, \lambda_{j_2}, \gamma, T\right)\frac{T^5}{hB}+\mathcal{O}\left(\frac{1}{B}\right).$$

The result in this Corollary is obtained by combining the above results. And we include the notebooks of all derivations in the supplementary material. □

### D.3 THE CASE WHEN $A$ IS A GENERAL STABLE MATRIX

**Lemma 3** (MSE when $A$ is a general stable matrix ). *Let $A$ be a stable $n \times n$ matrix with distinct eigenvalues $\lambda_1, \cdots, \lambda_m$ and corresponding multiplicities $q_1, \cdots, q_m$. There exist some constants $\{\bar{C}_i\}_{i=1}^m$, $\bar{C}_0$ and $C_j(\lambda_1, \cdots, \lambda_m, \gamma, T)$'s, such that the mean-squared error of the Monte-Carlo estimator in different setting satisfies*

*(1) Finite-Horizon undiscounted setting:*

$$MSE_T \in \big[\sum_{i=1}^m q_i\bar{C}_iMSE_T(h, B, \lambda_i), \quad C_1(\lambda_1, \cdots, \lambda_m, T)\sigma^4T^2h^2$$

$$+ \frac{\left(\bar{C}_2+C_3(\lambda_1, \cdots, \lambda_m, T)\mathcal{O}(T)\right)\sigma^4T^{2n+3}}{Bh}+\mathcal{O}(h^3)+\mathcal{O}(\frac{1}{B})\big], \qquad (39)$$

*where $MSE_T(h, B, \lambda_i)$ is the mean-squared error of the Monte-Carlo estimator in Theorem 1 by replacing the drift $a$ by $\lambda_i$.*

*(2) Finite-Horizon discounted setting:*

$$MSE_T \in \big[\sum_{i=1}^m q_i\bar{C}_iMSE_T(h, B, \gamma, \lambda_i), \quad C_4(\lambda_1, \cdots, \lambda_m, \gamma, T)\sigma^4\gamma^{2T}T^2h^2$$

$$+ C_5(\lambda_1, \cdots, \lambda_m, \gamma, T)\sigma^4\gamma^Th^3 + C_6(\lambda_1, \cdots, \lambda_m, T)\sigma^4h^4$$

$$+ \frac{\left(C_7(\lambda_1, \cdots, \lambda_m, \gamma, T)\right)\sigma^4T^{2n-1}}{Bh}+\mathcal{O}(h^5)+\mathcal{O}(\frac{1}{B})\big], \qquad (40)$$

*where $MSE_T(h, B, \gamma, \lambda_i)$ is the mean-squared error of the Monte-Carlo estimator in Lemma 2 by replacing the drift $a$ by $\lambda_i$.*

*(3) Infinite-Horizon discounted setting:*

$$MSE_\infty \in \big[\sum_{i=1}^m q_i\bar{C}_iMSE_\infty(h, B, \gamma, \lambda_i),$$

$$\left(C_8(\lambda_1, \cdots, \lambda_m, \gamma, T)+C_9(\lambda_1, \cdots, \lambda_m, \gamma, T)h\right)\sigma^4\gamma^{2T}$$

$$+ \left(C_{10}(\lambda_1, \cdots, \lambda_m, \gamma, T)h^2+C_{11}(\lambda_1, \cdots, \lambda_m, \gamma, T)h^3\right)\sigma^4\gamma^T$$

$$+ C_{12}(\lambda_1, \cdots, \lambda_m, T)\sigma^4h^4+\frac{\left(C_{13}(\lambda_1, \cdots, \lambda_m, \gamma, T)\right)\sigma^4T^{2n-1}}{Bh}$$

$$+ \mathcal{O}(h^5)+\mathcal{O}(\frac{1}{B})\big], \qquad (41)$$

*where $MSE_\infty(h, B, \gamma, \lambda_i)$ is the mean-squared error of the Monte-Carlo estimator in Theorem 3 by replacing the drift $a$ by $\lambda_i$.*

*Proof.* As we can see the proof of Lemma 2 is based on the proof of Theorem 1 with adding a discount factor $\gamma$, and the proof of Theorem 3 is based on the proof of Lemma 2 with the decomposition Eq. (17). By using the same flow direction, it is sufficient to show the result in case (1) and the results in case (2) and (3) follows.

Consider the decomposition of $\text{MSE}_T$ in finite-horizon undiscounted setting:

$$\text{MSE}_T = \mathbb{E}\left[(\hat{V}_M - V_T)^2\right]$$

$$= \mathbb{E}\left[\left(\hat{V}_M - \mathbb{E}\left[\hat{V}_M\right] + \mathbb{E}\left[\hat{V}_M\right] - V_T\right)^2\right]$$

$$= \underbrace{\mathbb{E}\left[\hat{V}_M^2\right] - \mathbb{E}\left[\hat{V}_M\right]^2}_{\text{Part1}} + \underbrace{\left(\mathbb{E}\left[\hat{V}_M\right] - V_T\right)^2}_{\text{Part2}}$$

Before the analysis of part 1 and part 2, we will introduce the following mean-squared error notations for the finite-horizon undiscounted scalar case with drift $\lambda_i$:

$$\text{MSE}_T(h, B, \lambda_i) = \text{Var}(h, \lambda_i) + \text{Approximation}(h, B, \lambda_i), \tag{42}$$

where $\text{Var}(h, \lambda_i) = \mathbb{E}\left[\hat{V}_M^2\right] - \mathbb{E}\left[\hat{V}_M\right]^2$ and $\text{Approximation}(h, B, \lambda_i) = \left(\mathbb{E}\left[\hat{V}_M\right] - V_T\right)^2$.

For part 1:

$$\mathbb{E}\left[\hat{V}_M^2\right] = \frac{h^2}{M} \sum_{i,j,k,l} \mathbb{E}\left[X_i(kh)^\top Q X_i(kh) X_j(lh)^\top Q X_j(lh)\right]$$

$$= \frac{h^2}{M^2} \sum_{i,j,k,l} \left[\mathbb{E}\left[X_i(kh)^\top Q X_i(kh)\right]\mathbb{E}\left[X_j(lh)^\top Q X_j(lh)\right] + 2\text{tr}\left\{Q\mathbb{E}\left[X_i(kh)X_j(lh)^\top\right]\right\}^2\right]$$

$$= h^2 \sum_{k,l} \mathbb{E}\left[X(kh)^\top Q X(kh)\right]\mathbb{E}\left[X_j(lh)^\top Q X(lh)\right] +$$

$$\frac{2h^2}{M} \sum_{k} \text{tr}\left\{Q\mathbb{E}\left[X(kh)X(kh)^\top\right]\right\}^2 + \frac{4h^2}{M} \sum_{k<l} \text{tr}\left\{Q\mathbb{E}\left[X(kh)X(lh)^\top\right]\right\}^2, \tag{43}$$

where the second equality is based on Isserlis' theorem and the trace operation.

Notice that $\mathbb{E}\left[\hat{V}_M\right]^2 = h^2 \sum_{k,l}\mathbb{E}\left[X(kh)^\top Q X(kh)\right]\mathbb{E}\left[X(lh)^\top Q X(lh)\right]$, thus

$$\mathbb{E}\left[\hat{V}_M^2\right] - \mathbb{E}\left[\hat{V}_M\right]^2$$

$$= \frac{2h^2}{M} \sum_{k} \text{tr}\left\{Q\mathbb{E}\left[X(kh)X(kh)^\top\right]\right\}^2 + \frac{4h^2}{M} \sum_{k<l} \text{tr}\left\{Q\mathbb{E}\left[X(kh)X(lh)^\top\right]\right\}^2$$

To analyze the above form, we decompose the matrix $A$ by it Jordan form, i.e. $A = P^{-1}JP$ for some inevitable matrix $P$ and $J = \text{diag}(J_i, \cdots, J_m)$, where $J_i$ is the Jordan block corresponding to the eigenvalue $\lambda_i$.

Notice that $e^{J(kh-s)} = \text{diag}(e^{J_1(kh-s)}, \cdots, e^{J_m(kh-s)})$, where

$$e^{J_i(kh-s)} = e^{\lambda_i(kh-s)}\begin{pmatrix} 1 & kh-s & \frac{(kh-s)^2}{2!} & \cdots & \frac{(kh-s)^{q_i-1}}{(q_i-1)!} \\ & 1 & kh-s & \cdots & \frac{(kh-s)^{q_i-2}}{(q_i-2)!} \\ & & \ddots & & \vdots \\ & & & & 1 \end{pmatrix}.$$

Combining with the fact that for any $k, l$,

$$\mathbb{E}\left[X(kh)X(lh)^\top\right] = \int_0^{kh \wedge lh} e^{A(kh-s)} e^{A^\top(lh-s)}\,\mathrm{d}s$$

$$= \int_0^{kh \wedge lh} P^{-1}e^{J(kh-s)}PP^\top e^{J^\top(lh-s)}P^{-\top}\,\mathrm{d}s,$$

we can conclude that for any $k \leq l$, $\text{tr}\left\{Q\mathbb{E}\left[X(kh)X(lh)^\top\right]\right\}$ is a linear combination of $\mathcal{L}_{1,i,j}$ and $\mathcal{L}_{2,i,j}$ for all $i, j$, where

$$\mathcal{L}_{1,i,j} := C_{1,i,j} \int_0^{kh} e^{(\lambda_i(kh-s)+\lambda_j(lh-s))} \, \mathrm{d}s$$

$$= C_{1,i,j} \frac{e^{\lambda_i kh} + e^{\lambda_j}}{\lambda_i + \lambda_j} \left(1 - e^{-(\lambda_i+\lambda_j)kh}\right)$$

$$\mathcal{L}_{2,i,j} := C_{2,i,j} \int_0^{kh} e^{(\lambda_i(kh-s)+\lambda_j(lh-s))} (kh-s)^{\tilde{q}_i}(lh-s)^{\tilde{q}_j} \, \mathrm{d}s \,,$$

where $C_{1,i,j}, C_{i,j}$ are some constants and $\tilde{q}_i \in \{0, \cdots, q_i - 1\}$, $\tilde{q}_j \in \{0, \cdots, q_j - 1\}$.

For the integral in $\mathcal{L}_{2,i,j}$, as $\tilde{q}_i + \tilde{q}_j \leq n - 1$, we can have the inequality:

$$\int_0^{kh} e^{(\lambda_i(kh-s)+\lambda_j(lh-s))}(kh-s)^{\tilde{q}_i}(lh-s)^{\tilde{q}_j} \, \mathrm{d}s$$

$$\leq T^{n-1} \int_0^{kh} e^{(\lambda_i(kh-s)+\lambda_j(lh-s))} \, \mathrm{d}s \,. \tag{44}$$

Since

$$\text{tr}\left\{Q\mathbb{E}\left[X(kh)X(lh)^\top\right]\right\}^2 = \sum_{i_1,j_1,i_2,j_2} \sum_{k,l} \prod_{l_1,l_2 \in \{1,2\}} \mathcal{L}_{l_1,i_1,j_1}\mathcal{L}_{l_2,i_2,j_2} \,,$$

and all the terms are nonnegative. We drop all terms that include $\mathcal{L}_{2,i,j}$ factor and only include the $\mathcal{L}_{1,i,i}^2$ with $k = l$ terms in the lower bound of part 1. That is to say, the lower bound of part 1 is $\sum_{i=1}^m q_i \bar{C}_i \text{Var}(h, \lambda_i)$.

The upper bound of part 1 can be obtained by replacing all $\mathcal{L}_{1,i,j}$ factors by $\mathcal{L}_{2,i,j}$ and use the bound given in Eq. (44). This leads to the upper bound for part 1 is $\frac{(\bar{C}_2 + C_3(\lambda_1, \cdots, \lambda_m, T)\mathcal{O}(T))\sigma^4 T^{2n+5}}{Bh} + \mathcal{O}(\frac{1}{B})$.

For Part 2, let $g(t) = \mathbb{E}\left[X(t)^\top Q X(t)\right]$ on $[0, T]$. Then $\mathbb{E}\left[\hat{V}_M\right]$ is the left Riemann sum approximation of $g(t)$, by the property of Riemann approximation,

$$|\mathbb{E}\left[\hat{V}_M\right] - V_T| \approx 2hTg(T) + \mathcal{O}(h^2) \,,$$

where

$$g(T) = \text{tr}\left\{Q\mathbb{E}\left[X(T)X(T)^\top\right]\right\} = \sigma^2 \text{tr}\left\{Q \int_0^T e^{A(t-s)}e^{A^\top(t-s)} \, \mathrm{d}s\right\} \,,$$

which is a constant depends on $\lambda_1, \cdots, \lambda_m, T$. Thus

$$(\mathbb{E}\left[\hat{V}_M\right] - V_T)^2 \approx C_1(\lambda_1, \cdots, \lambda_m, T)\sigma^4 T^2 h^2 + \mathcal{O}(h^3) \,,$$

which has the same order in $h$ as the scalar case in finite-horizon undiscounted setting. Thus the result in Eq. (39) is obtained by combining part 1 bounds and part 2 approximation.

As we explained in the beginning of this proof, in the finite-horizon discounted setting, we will follow the similar arguments as in the proof of Eq. (39) to obtain result Eq. (40).

For the infinite-horizon discounted setting, the corresponding part 1 in the $\text{MSE}_\infty$ is the same as the part 1 in $\text{MSE}_T$ of Eq. (40). The part 2 is approximated by using the decomposition Eq. (17) and the fact that

$$V_{t,\infty} = \int_T^\infty \gamma^t \mathbb{E}\left[X(t)^\top Q X(t)\right] dt = \gamma^T C(r, T, \lambda_1, \cdots, \lambda_m) \,.$$

To verify $V_{T,\infty}$ is $\mathcal{O}(\gamma^T)$, one can find the bounds of $V_{T,\infty}$ by using the similar arguments in the above proof of Eq. (39) and the following inequality:

$$\int_0^t e^{(\lambda_i+\lambda_j)(t-s)}(t-s)^{\tilde{q}_i+\tilde{q}_j}\,\mathrm{d}s$$

$$\leq t^{n-1}\int_0^t e^{(\lambda_i+\lambda_j)(t-s)}\,\mathrm{d}s = \frac{t^{n-1}}{(\lambda_i+\lambda_j)}\left(e^{(\lambda_i+\lambda_j)t}-1\right).$$

Then the components in $\int_T^\infty \gamma^t \mathbb{E}\left[X(t)X(t)^\top\right]\,\mathrm{d}t$ is lower bounded by $\int_T^\infty \frac{r^t}{(\lambda_i+\lambda_j)}\left(e^{(\lambda_i+\lambda_j)t}-1\right)\,\mathrm{d}t$ and upper bounded by $\int_T^\infty \frac{r^t t^{n-1}}{(\lambda_i+\lambda_j)}\left(e^{(\lambda_i+\lambda_j)t}-1\right)\,\mathrm{d}t$. By the celebrating approximation of incomplete gamma function when $T$ is large, we have

$$\int_T^\infty \frac{r^t t^{n-1}}{(\lambda_i+\lambda_j)}\left(e^{(\lambda_i+\lambda_j)t}-1\right)\,\mathrm{d}t \approx \frac{r^T T^{n-1}}{(\lambda_i+\lambda_j)}\left(e^{(\lambda_i+\lambda_j)T}-1\right).$$

Followed by

$$V_{T,\infty} = \mathrm{tr}\left\{Q\int_T^\infty \gamma^t \mathbb{E}\left[X(t)X(t)^\top\right]\,\mathrm{d}t\right\},$$

we can obtain that $V_{T,\infty} = \gamma^T C(r, T, \lambda_1, \cdots, \lambda_m)$.

This result leads to the fact that part 2 is

$$\left(C_8(\lambda_1,\cdots,\lambda_m,\gamma,T) + C_9(\lambda_1,\cdots,\lambda_m,\gamma,T)h\right)\sigma^4\gamma^{2T} + \left(C_{10}(\lambda_1,\cdots,\lambda_m,\gamma,T)h^2\right.$$

$$\left. +C_{11}(\lambda_1,\cdots,\lambda_m,\gamma,T)h^3\right)\sigma^4\gamma^T + C_{12}(\lambda_1,\cdots,\lambda_m,T)\sigma^4 h^4 + \mathcal{O}(h^5),$$

which coincides with the $\mathrm{Var}(h\lambda_i)$ in the infinite-horizon discounted scalar case. The results in (3) then follows. $\qquad\square$

## E  TRADE-OFF IN LQR WITH DENSE DIAGONALIZABLE MATRICES

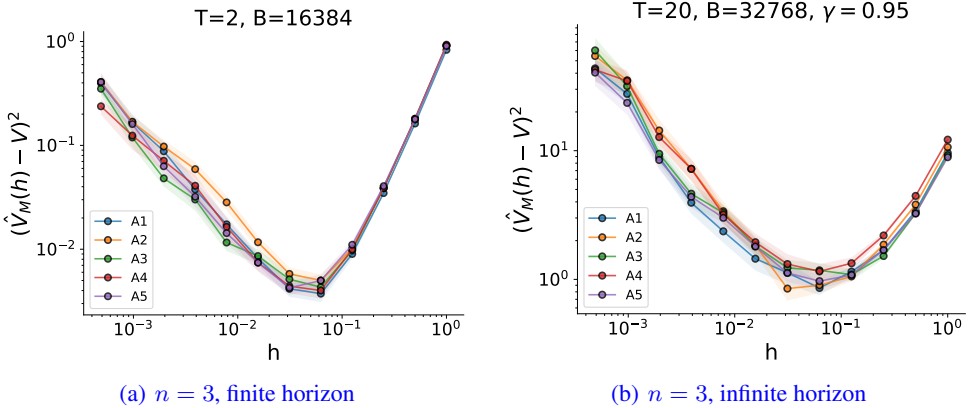

(a) $n = 3$, finite horizon  (b) $n = 3$, infinite horizon

Figure 4: Mean-squared error trade-off in LQR with random symmetric diagonalizable drift matrices A. The matrices A{1,2,3,4,5} in each plot were generated by randomly sampling an eigendecomposition and then using it to compute the drift matrix $A$. The eigenvalues are uniformly sampled on bounded disjoint intervals and the eigenvectors are sampled randomly from the classical compact groups detailed in Mezzadri (2006). Note the matrices in (a) and (b) are not equal.

Recall that in Fig. 1(b) and Fig. 1(c), the drift matrices $A$ were scaled identity matrices. In this section, we show empirically that the trade-off persists when $A$ is a dense, stable matrix which aligns with our theoretical results Theorem 2 and Corollary 4. Fig. 4 clearly shows a trade-off for 10 randomly sampled $3 \times 3$ dense, stable matrices. The procedure for randomly sampling a systems starts with uniformly sampling two eigenvalues from disjoint, bounded intervals, $\lambda_1 \in [-1.5, -1.0)$ and $\lambda_3 \in (-1.0, -0.75]$. The final eigenvalue is set to be $\lambda_2 = -1.0$. Note that since all the

eigenvalues sampled are negative, any matrix whose eigenvalues are $\lambda_1, \lambda_2, \lambda_3$ is said to be stable. Next, we randomly sample an orthogonal matrix $Q$ using a built-in SCIPY (Virtanen et al., 2020) routine, ORTHO_GROUP.RVS. Now let $\Lambda = \text{diag}(\lambda_1, \lambda_2, \lambda_3)$, the random, dense, stable matrix $A$ is defined as $A = Q^\top \Lambda Q$.

## F  EXPERIMENTAL DETAILS

We summarize the environment-specific parameters in Table 1 for the nonlinear-system experiments.

Table 1: The setup of the environments. *: In MuJoCo environments in OpenAI Gym, $\delta t = $ timestep $*$ frame_skip. $\delta t = 0.001$ seconds for the proxies to the continuous-time environments. Note that 'timestep' (the step size of the MuJoCo dynamics simulation) and 'frame_skip' (the algorithmic step size) are two quantities in the implementation of OpenAI Gym MuJoCo environments.

| Environment | Episode Length (steps) | Original* $\delta t$ | Horizon $T$ (seconds) |
|---|---|---|---|
| Pendulum | 200 | 0.05 | 10 |
| BipedalWalker | 500 | 0.05 | 10 |
| InvertedDoublePendulum | 1000 | 0.05 | 50 |
| Pusher | 1000 | 0.05 | 50 |
| Swimmer | 1000 | 0.04 | 40 |
| Hopper | 1000 | 0.008 | 8 |
| HalfCheetah | 1000 | 0.05 | 50 |
| Ant | 1000 | 0.05 | 50 |

| Environment | $B_0$ | $h$ |
|---|---|---|
| Pendulum | $10k$ | [0.001, 0.002, 0.004, 0.01, 0.02, 0.04, 0.1] |
| BipedalWalker | $10k$ | [0.001, 0.002, 0.004, 0.01, 0.02, 0.04, 0.1] |
| InvertedDoublePendulum | $25k$ | [0.002, 0.004, 0.01, 0.02, 0.04, 0.1, 0.2, 0.4, 1] |
| Pusher | $25k$ | [0.002, 0.004, 0.01, 0.02, 0.04, 0.1, 0.2, 0.4, 1] |
| Swimmer | $20k$ | [0.002, 0.004, 0.01, 0.02, 0.04, 0.1] |
| Hopper | $8k$ | [0.001, 0.002, 0.004, 0.01, 0.02, 0.04, 0.1] |
| HalfCheetah | $25k$ | [0.002, 0.004, 0.01, 0.02, 0.04, 0.1, 0.2, 0.4] |
| Ant | $25k$ | [0.002, 0.004, 0.01, 0.02, 0.04, 0.1, 0.2, 0.4] |

