# OpenReview forum: "Managing Temporal Resolution in Continuous Value Estimation: A Fundamental Trade-off"
_ICLR.cc/2023/Conference — Submitted to ICLR 2023_

### Official Review · Reviewer_pVUY · 2022-10-24

**Confidence:** 3
**Correctness:** 3
**Technical Novelty And Significance:** 2
**Empirical Novelty And Significance:** 2
**Recommendation:** 3

**Clarity, Quality, Novelty And Reproducibility:**

The paper is easy to follow. However, as I discussed for the previous question, the novelty of this work is not discussed sufficiently, especially for the main proof techniques.

**Strength And Weaknesses:**

Strength:

This paper has a very good motivation to study the impact of discretization for general value estimation in reinforcement learning (RL) and optimal control. The main theoretical results are presented clearly with the discussion of their implications.

Weakness:

1.  It is hard to see the broader impact of this work. The problem setting seems to be restrictive: The results in this work only applies time-invariant LQR system, state-feedback controllers, and a specific form of disturbances. To address this concern, I encourage the authors to be more specific about what the “implications” are for more general settings. For example, if one wants to optimize the policy to minimize the continuous-time objective, how should the temporal discretization be chosen? Do the optimal temporal discretization depend on the current policy?
2. The discussion about the technical difficulty for deriving the main result is not sufficient in the main body. As the authors discussed below Corollary 1, the error terms can be understood as a Riemann sum approximation error and the variance term. Before this work, are there existing results that bound Riemann sum approximation error? I believe this is important to clarify, because I feel bounding the variance term (It is usually called the stochastic error in RL literature) via independent trials is a standard technique.
3. For the numerical simulations, I recommend the authors to do a more careful comparison with the theoretical optimal temporal discretization $h$, because I feel the trend of bias-variance trade-off can be expected even without the theoretical analysis in this work. It is more important to see how well the actual optimal $h$ matches the theoretical prediction $h^*(B)$.


**Summary Of The Paper:**

This paper studies the how discretization over the time horizon affect the error of LQR value estimation compared with the true continuous-time LQR value. The main result is about the mean squared error (MSE) of the Monte-Carlo policy evaluation, which reveals a trade-off between discretization time interval $h$ and the number of trajectories $M$. When the budget of total sample points is fixed, one can use this bound to find the optimal discretization time interval $h$ to minimize the MSE. Lastly, the authors use numerical simulations to demonstrate their theoretical findings in both LQR settings and nonlinear settings.

**Summary Of The Review:**

This paper studies a relatively restrictive problem setting, and it is unclear whether the results or the proof techniques here apply to more general settings. Besides, I feel the technical contributions are not significant enough, and the numerical simulations are not sufficient to verify the theoretical findings quantitatively. Therefore, I recommend for reject.

---

> ### Author Response · Authors · 2022-11-16
> **Response to Reviewer pVUY (Part 1)**
>
> Dear Reviewer:
>
> Thank you for your time reviewing the paper and for your thoughtful feedback. Below please find our responses to your questions and comments:
>
> > It is hard to see the broader impact of this work. The problem setting seems to be restrictive: The results in this work only applies time-invariant LQR system, state-feedback controllers, and a specific form of disturbances.
>
> `Re:` Although the exact analysis in this paper addresses a time-invariant LQR system, the implications of this work are far broader.  In our submission, we present a significant effort that shows how the trade-off persists even in non-linear systems. We further added more detailed numerical evidence that the empirically observed value of $h^*$ aligns well with the theoretically predicted trade-off, even in the non-linear setting (see Figure 3 in the updated manuscript). More discussion on this below.
>
> We disagree with the sentiment that studying simple systems (time-invariant LQR in our case) does not have broader implications. First of all, studying special cases allows us to precisely understand a trade-off that had not been previously identified in the literature, which might be significantly more difficult or impossible in this form for general systems. The linear analysis also provides a starting point for extending the analysis to non-linear systems via local linearization arguments, which we believe is an interesting question for future work.
>
> Historically, there is a rich history of work done [1,2,3,4,5,6,7,8] in time-invariant LQR systems and state-feedback controllers with specific forms of disturbance that have heavily influenced works done in more general settings [9,10,11,12,13,14,15,16,17,18]. Note that [9,10] were heavily influenced by results shown in [1], [11] was influenced by results shown in [2],  [12] was heavily influenced by results shown in [3], [13] borrow insights and analysis from [4], [14] builds off of the work done by [5], [15,16] were influenced by work done in [7], [17] generalizes the results shown in [7], while [18] generalizes the results shown in [8]. We can include the above discussion in the related works section of our paper.
>
> References:
>
> [1] Tu, Stephen and Benjamin Recht. “The Gap Between Model-Based and Model-Free Methods on the Linear Quadratic Regulator: An Asymptotic Viewpoint.” COLT (2019).
> [2] Abbasi-Yadkori, Yasin and Csaba Szepesvari. “Regret Bounds for the Adaptive Control of Linear Quadratic Systems.” COLT (2011).
> [3] Dean, Sarah et al. “On the Sample Complexity of the Linear Quadratic Regulator.” ArXivabs/1710.01688 (2020).
> [4] Simchowitz, Max and Dylan J. Foster. “Naive Exploration is Optimal for Online LQR.” ICML(2020).
> [5] Dean, Sarah et al. “Regret Bounds for Robust Adaptive Control of the Linear Quadratic Regulator.” NeurIPS (2018).
> [6] Fiechter, Claude-Nicolas. “PAC adaptive control of linear systems.” COLT '97 (1997).
> [7] Fazel, Maryam et al. “Global Convergence of Policy Gradient Methods for the Linear Quadratic Regulator.” ICML(2018).
> [8] Malik, Dhruv et al. “Derivative-Free Methods for Policy Optimization: Guarantees for Linear Quadratic Systems.” J. Mach. Learn. Res. 21 (2019): 21:1-21:51.
> [9] Xu, Tengyu, Zhe Wang, and Yingbin Liang. "Improving sample complexity bounds for (natural) actor-critic algorithms." NeurIPS (2020).
> [10] Sun, Wen, Nan Jiang, Akshay Krishnamurthy, Alekh Agarwal, and John Langford. "Model-based rl in contextual decision processes: Pac bounds and exponential improvements over model-free approaches." COLT (2019).
> [11] Gradu, Paula et al. “Adaptive Regret for Control of Time-Varying Dynamics.” ArXivabs/2007.04393 (2020).
> [12] Xu, Huazhe et al. “Algorithmic Framework for Model-based Reinforcement Learning with Theoretical Guarantees.” ArXiv abs/1807.03858 (2019).
> [13] Kakade, Sham M. et al. “Information Theoretic Regret Bounds for Online Nonlinear Control.” ArXiv abs/2006.12466 (2020).
> [14] Lale, Sahin et al. “Reinforcement Learning with Fast Stabilization in Linear Dynamical Systems.” AISTATS (2022).
> [15] Walsh, Thomas J. et al. “Exploring compact reinforcement-learning representations with linear regression.” UAI (2009).
> [16] Strehl, Alexander L. and Michael L. Littman. “Online Linear Regression and Its Application to Model-Based Reinforcement Learning.” NIPS (2007).
> [17] Bhandari, Jalaj and Daniel Russo. “Global Optimality Guarantees For Policy Gradient Methods.” ArXiv abs/1906.01786 (2019).
> [18] Wang, Lingxiao et al. “Neural Policy Gradient Methods: Global Optimality and Rates of Convergence.” ArXiv abs/1909.01150 (2020).

---

> > ### Author Response · Authors · 2022-11-16
> > **Response to Reviewer pVUY (Part 2)**
> >
> > > To address this concern, I encourage the authors to be more specific about what the “implications” are for more general settings. For example, if one wants to optimize the policy to minimize the continuous-time objective, how should the temporal discretization be chosen? Do the optimal temporal discretization depend on the current policy?
> >
> > `Re:` There are several important implications of the presented results for usage in practice:
> >
> > - The first message is that a carefully chosen step-size can lead to better estimation in policy evaluation. All libraries that we are aware of use hard-coded values for the step-size. This is a clear indication that previously there was little attention to choosing the time discretization, and our results show that this can lead to inefficient data usage.
> > - Both theoretical and empirical results suggest that the dependence of the MSE on the step-size exhibits a U-shaped / convex behavior.
> > - Both theoretical and empirical results confirm that the optimal step-size depends on the environment / policy, as expected.
> > - The predicted dependence of the optimal step size on the data budget (and the horizon) is also observed in the non-linear setting (see Figure 3 in the updated manuscript). Note that this gives a reasonable starting point for optimizing the step-size: $h^* = (0.1 \pm 0.08) * TB^{-1/3}$.
> > - The latter finding further has important implications for an “upscaling” scenario, where the step-size is first optimized as a hyperparameter for a smaller budget, and then the predicted dependence is used to compute a good step-size for a larger data budget.
> >
> > Given the evidence presented above, an interesting direction for future research is to extend the results to more general non-linear systems, e.g. via local linearization arguments or using closed-form solutions for stochastic processes.
> >
> > > The discussion about the technical difficulty for deriving the main result is not sufficient in the main body. As the authors discussed below Corollary 1, the error terms can be understood as a Riemann sum approximation error and the variance term. Before this work, are there existing results that bound Riemann sum approximation error? I believe this is important to clarify, because I feel bounding the variance term (It is usually called the stochastic error in RL literature) via independent trials is a standard technique.
> >
> > `Re:` The technical contribution of this paper is an *exact* expression of the mean-squared error of the Monte-Carlo estimator. Note that this is different from merely *bounding* the Riemann sum estimation error, as the exact result verifies the *tightness* of the claimed trade-off. Obtaining the trade-off involves the following non-trivial technical steps: i) calculating fourth-order cross moments of the Langevin process, ii) obtaining closed-form expressions for exact and Monte-Carlo approximation of the values, iii) computing an exact expression of the mean-square error, iv) optimizing the trade-off between the number of episodes and the step-size for a fixed data budget, v) understanding the exact trade-off through Taylor approximations in the most relevant regime, and vi) extending the analysis to multi-dimensional systems.
> >
> > We do not claim that the technical analysis represents the main contribution of this work. However, to the best of our knowledge, this is the first time that a time-discretization trade-off has been identified and analyzed in LQR systems, and our results show nuanced trade-offs that exhibit different behavior depending on discounting and horizon. For example in the undiscounted setting the optimal step size scales approximately like $B^{-1/3}$. In the discounted setting, the optimal choice for the step size scales approximately like $B^{-1/5}$. We claim this finding is neither obvious nor previously known.  Even though the existence of a trade-off might appear intuitive in retrospect, it has not been previously reported and our results add a significant level of detail to such a claim. Formalizing the characterization of the trade-off (due to having finite interactions with a continuous time system) was not obvious, let alone obtaining an exact expression for the trade-off.
> >
> > Moreover, the paper goes beyond the linear setting by verifying the results in several nonlinear experiments. As mentioned above, additional analysis highlights that the predicted trade-off is observed in these settings despite the assumptions not being satisfied (Figure 3 in the updated manuscript).

---

> > > ### Author Response · Authors · 2022-11-16
> > > **Response to Reviewer pVUY (Part 3)**
> > >
> > > > For the numerical simulations, I recommend the authors to do a more careful comparison with the theoretical optimal temporal discretization h, because I feel the trend of bias-variance trade-off can be expected even without the theoretical analysis in this work. It is more important to see how well the actual optimal h matches the theoretical prediction h∗(B)
> > >
> > > `Re`: We thank the reviewer for the suggestion to more thoroughly compare the numerical results to the theoretical trade-off. We added Fig 3 in the updated manuscript, that shows the empirical $h^*$ as a function of $B$ and $T$ for all nonlinear systems experiments. It strongly suggests that the findings in the linear setting do in fact extend to the nonlinear setting.
> > >
> > > We're not aware that the trade-off has been studied before. As detailed above, the results go well beyond simply observing a trend in the trade-off; we provide precise and nuanced results that characterize the trade-off exactly.
> > >
> > > As far as we know, optimizing the trade-off has been mostly neglected in past empirical work (e.g. popular libraries like OpenAI Gym and Deepmind Control Suite do not provide the step-size as input variable). There were a few works that gave empirical evidence of the benefit of managing temporal resolution as we reviewed in the related works section, but none of them characterized the trade-off.

---

> ### Author Response · Authors · 2022-11-25
> **Follow up to Reviewer pVUY**
>
> We wanted to follow up to see if our responses addressed your concerns or if further elaboration is needed. Thank you again!

---

> ### Comment · Reviewer_pVUY · 2022-12-01
> **Thanks for the detailed response!**
>
> I want to thank the authors for the detailed response! However, I would like to clarify one of my major concern if it is not clear from my original review: In a digital control system where the control input $u(t)$ can only be updated every $\delta t$ seconds, does it make sense to down sample $h$ (i.e., make it larger than $\delta t$) under a fixed data budget $B$? Since the controller needs to observe the state to decide the control input every $\delta t$ seconds, it does not require much additional effort for the controller to also record its state and control input every $\delta t$ seconds to compute the accumulative cost. I feel the fixed data budget only makes sense if the controller is also implemented in continuous time, but it happens rarely in practice.

---

> > ### Author Response · Authors · 2022-12-08
> > **Response to the additional concern**
> >
> > Dear Reviewer pVUY,
> >
> > Thank you for your additional comments.
> >
> > To address the new concern that you brought forth regarding the scope of our work, our analysis was carried out directly on the closed-loop dynamics assuming a continuous-time controller, given by static feedback of the state. This was stated in and above Eq. (3) in our paper. As we discuss in the paper, downsampling of the closed-loop dynamics is useful to reduce processing and storage requirements, and can be implemented in many scenarios including numerical simulation as we demonstrate in our experiments. We will be happy to further elaborate if needed.

---

### Official Review · Reviewer_UU8T · 2022-10-28

**Confidence:** 3
**Correctness:** 4
**Technical Novelty And Significance:** 3
**Empirical Novelty And Significance:** 3
**Recommendation:** 6

**Clarity, Quality, Novelty And Reproducibility:**

The paper is well-written and The claim drawn from previous theoretical analysis is well supported by the empirical results.

**Strength And Weaknesses:**

**Strength**
- The one dimension example presented in the paper is helpful for understanding
- The experiments in the paper clearly support the theoretical analysis and results.


**Weakness & Questions**

- For the non-linear case, why you choose a very short horizon. What happens if you choose $T=1000$ which is default in mujoco, and $\gamma = 0.001$ ? Will you get similar results here?

- For practise usage, it seems that we still don't know how to choose the optimal time discretization. Do you have any guidance or idea how to choose that in practise?

**Summary Of The Paper:**

This paper examines the time discretization for continuous value estimation. By analyzing Monte-Carlo value estimation for LQR systems for both finite-horizon and infinite discounted horizon settings, the authors finds that there is a fundamental trade-off between approximation error and statistical error in value estimation, which indicates there is an optimal choice for time discretization that depends on the data budget. The authors also demonstrate the trade-off in numerical simulation of LQR instances and non-linear mujoco environments.

**Summary Of The Review:**

Overall I think this is an interesting paper, which shows a trade-off for time discretization choice, which could be a potentially important guidance for practical usage.

---

> ### Author Response · Authors · 2022-11-16
> **Response to Reviewer UU8T**
>
> Dear Reviewer:
>
> Thank you for your time reviewing the paper and for your thoughtful feedback. Below please find our responses to your questions and comments:
>
> > For the non-linear case, why do you choose a very short horizon? What happens if you choose T=1000 which is default in mujoco, and γ=0.001? Will you get similar results here?
>
> `Re:` The horizon that we used for MuJoCo experiments matches the original setting of $1000$ steps. The confusion may have arisen from the unit of physical time $T$. Since we aimed to understand the discretization of continuous time systems, $T$ in our Fig.2 was in physical time (seconds) rather than in the number of discrete steps (noted in the caption of Fig.2). For instance, in Swimmer-v3, 1000 steps under the original $\delta t = 0.04$ in OpenAI Gym translates to $T=40$ (seconds), which is the physical time horizon that we used in our experiment for all choices of $h$. We've updated the paper (Sec. 4.2 and Appendix F) with additional clarifications (shown in blue).
>
> As to the question about $\gamma=0.001$, the nonlinear systems experiments are in the undiscounted setting ($\gamma=1$) as noted in the paper (above Fig.2). We believe that $\gamma=1$ is the common setting for these environments.
>
> > For practise usage, it seems that we still don't know how to choose the optimal time discretization. Do you have any guidance or idea how to choose that in practice?
>
> `Re:` There are several important implications of our results for usage in practice:
>
> - The first message is that a carefully chosen step-size can lead to better estimation in policy evaluation. All libraries that we are aware of use hard-coded values for the step-size. This is a clear indication that previously there was little attention to choosing the time discretization, and our results show that this can lead to inefficient data usage.
> - Both theoretical and empirical results suggest that the dependence of the MSE on the step-size exhibits a U-shaped / convex behavior.
> - Both theoretical and empirical results confirm that the optimal step-size depends on the environment / policy, as expected.
> - The predicted dependence of the optimal step size on the data budget (and the horizon) is also observed in the non-linear setting (see Figure 3 in the updated manuscript). Note that this gives a reasonable starting point for optimizing the step-size: $h^* = (0.1 \pm 0.08)* T B^{-1/3}$.
> - The latter finding further has important implications for an “upscaling” scenario, where the step-size is first optimized as a hyperparameter for a smaller budget, and then the predicted dependence is used to compute a good step-size for a larger data budget.

---

> ### Author Response · Authors · 2022-11-25
> **Follow up to Reviewer UU8T**
>
> We wanted to follow up to see if our responses addressed your concerns or if further elaboration is needed. Thank you again!

---

### Official Review · Reviewer_GwC4 · 2022-11-05

**Confidence:** 3
**Correctness:** 3
**Technical Novelty And Significance:** 3
**Empirical Novelty And Significance:** 3
**Recommendation:** 5

**Clarity, Quality, Novelty And Reproducibility:**

The paper is well-written and easy to follow, but some improvements are needed. See weaknesses and the following questions for details.

**Strength And Weaknesses:**

Strengths:
1. 	The problem considered in this paper is important, and the results provide insights for practitioners on how to choose suitable stepsize for approximating continuous systems with discrete measurements.
2. 	The linear system considered in this paper is typical, and the theoretical results are rich and complete, with a clear explanation.
3. 	The numerical results match the theory well.

Weaknesses
1. 	Corollary 2 calculates the optimal step size using a Taylor expansion and omits high-order terms w.r.t. $h$. However, such estimation is only correct when $h$ is small, and the approximation error of $h^{*}$ is not given in Corollary 2. Therefore, we do not know whether the estimation is good and precise enough.
2. 	In the numerical experiments, the author considered a three-dimensional linear system. However, they only consider several special cases where $A=cI$, which is too special and too similar to the 1-D case. I think considering more general cases will provide more insights for the high-dimensional case. For example, $A=\diag\{c_1, c_2, c_3\}$ where $c_1>c_2>c_3$.

**Summary Of The Paper:**

This paper studies the problem of approximating a continuous system from discrete measurements with a finite data budget. They first consider the simplest and canonical case of Monte-Carlo value estimation in a Langevin dynamical system with quadratic instantaneous costs. They obtain analytical expressions of the least-squares error that exactly characterize the approximation-estimation trade-off with respect to the step-size parameter. Second, they present a numerical study that illustrates and confirms the trade-off in both linear and non-linear systems, including several MuJoCo control environments. The findings imply that practitioners should pay attention to carefully choosing the step-size parameter of the estimation to obtain the most accurate results possible.

**Summary Of The Review:**

Additional Problems:

The problem considered in this paper is closely related to stochastic differential equations. Specifically, using discrete measurements to approximate SDEs is similar to using the numerical solution to solve SDEs (for example, Euler-Maruyama method). What is the difference between your findings and those traditional results in numerical SDE?

The theoretical results seem to be restricted to the linear case, can authors make more discussions on the difficulty of extending results to nonlinear cases?

---

> ### Author Response · Authors · 2022-11-16
> **Response to Reviewer GwC4**
>
> Dear Reviewer:
>
> Thank you for your time reviewing the paper and for your thoughtful feedback. Below please find our responses to your questions and comments:
>
> > Corollary 2 calculates the optimal step size using a Taylor expansion and omits high-order terms w.r.t. $h$. However, such estimation is only correct when $h$ is small, and the approximation error of $h^{*}$ is not given in Corollary 2. Therefore, we do not know whether the estimation is good and precise enough.
>
> `Re:` We first note that $h^*$ can be obtained **exactly** by minimizing the expression from Theorem 1 over all possible step sizes $h_m = T/m$ for $m = 1, \dots B$. This is stated below Theorem 1. As such, the main intention of Corollary 2 is to provide intuition of the behavior of the optimal step size w.r.t. the data budget, in a regime where the budget is large (hence the step size is small). We believe that this is the most relevant regime, as modern RL setups typically use a considerable amount of data. We have also added additional evidence that the predicted trade-off can be observed even in the non-linear experiments (see Fig 3 in the updated version of the manuscript).
>
> That said, we added a more precise description of h* to Appendix C.2 in the updated manuscript, which included higher-order terms of 1/B.
>
> > In the numerical experiments, the author considered a three-dimensional linear system. However, they only consider several special cases where $A=cI$, which is too special and too similar to the 1-D case. I think considering more general cases will provide more insights for the high-dimensional case. For example, $A=diag(c_1, c_2, c_3)$ where $c_1>c_2>c_3$.
>
> `Re:` Thank you for the valuable suggestion. We added experiments with randomly generated dense diagonalizable matrices (see Appendix E in the updated manuscript for the figure and more details). As before, we observe the trade-off in MSE in all systems in both finite horizon and infinite horizon settings.
>
> > The problem considered in this paper is closely related to stochastic differential equations. Specifically, using discrete measurements to approximate SDEs is similar to using the numerical solution to solve SDEs (for example, Euler-Maruyama method). What is the difference between your findings and those traditional results in numerical SDE?
>
> `Re:` Numerical solutions to SDEs deal with approximating the system evolution over time. In our case, the exact system evolution is sub-sampled through the time-discretization. And the SDE has an implicit solution, which is an O-U process. The MSE is only measuring the approximation error and variance of the Monte-Carlo estimator. Numerical solutions are not necessary. Furthermore, a numerical solution may cause more errors (even a strong convergent solution would always have numerical error) in the MSE result. Such numerical error may perturb the analysis of MSE.
>
> > The theoretical results seem to be restricted to the linear case, can authors make more discussions on the difficulty of extending results to nonlinear cases?
>
> `Re:` There are several possible directions for extending the results to non-linear cases, although we believe that these go beyond the scope of the current presentation.
>
> For nonlinear SDEs that are analytically solvable (e.g. reducible case or some other nonlinear forms in [1]), an exact or approximate expression for MSE can be obtained in a similar manner to this paper. However, many nonlinear SDEs do not have analytic solutions. One possible approach could be to locally linearize the system, in which case our results provide an important starting point for further analysis. A further question that arises in the non-linear setting is how the model complexity (e.g. the number of parameters) affects the trade-off.
> We also emphasize that we specifically chose the canonical LQR system in order to exactly characterize and study the presented trade-offs, which in this form might be rather challenging for other non-linear systems. Nevertheless, our empirical results on non-linear systems show that the theoretical predictions align with the experiments (see the new Figure 3 that shows the empirical dependence of $h^*$ on the data budget B aligns with the theoretical prediction).
>
> Reference:
>
> [1] Øksendal, Bernt. "Stochastic differential equations." Stochastic differential equations. Springer, Berlin, Heidelberg, 2003. 65-84

---

> ### Author Response · Authors · 2022-11-25
> **Follow up to Reviewer GwC4**
>
> We wanted to follow up to see if our responses addressed your concerns or if further elaboration is needed. Thank you again!

---

> > ### Comment · Reviewer_GwC4 · 2022-12-06
> > **Thanks for your response**
> >
> > Thanks for your detailed response. Can the authors provide more explanations on why the truncation at $O(h^3)$ is reasonable (in Eq (30))? If it is reasonable, how does the truncation affect the precision of the solution $h^*$? Can the authors provide more analytical results on the approximation error of $h^{*}$?

---

> > > ### Author Response · Authors · 2022-12-10
> > > **Response to the follow up question**
> > >
> > > Thank you for your additional feedback.
> > >
> > > A precise analysis of the approximation error for $h^*$ is quite complex; however, the exact $h^*$ can be obtained numerically (recall that it is difficult to analytically solve the optimization problem in Theorem 1).
> > > To better answer your question about the precision of $h^*$, we plotted the error of the approximate $h^*$ in Corollary 2 (compared to the exact $h^*$ that we obtained numerically) and share it on the following anonymized webpage:  https://sites.google.com/view/hstar-approx/home
> > > For the system instance (T=1, a=-1), the absolute error and the relative error both decrease as $B$ gets larger. At $B=$1e7, the absolute error is 6.521e-6, and the relative error is 0.00148.
> > > We find that such an approximation is reasonable for providing intuition about the scaling of $h^*$.
> > >
> > > We will be happy to further elaborate if needed.

---

### Decision · Program_Chairs · 2023-01-20

**Decision:**

Reject

**Justification For Why Not Higher Score:**

Concerns about the motivation and the simplicity of the theoretical model.

**Justification For Why Not Lower Score:**

N/A

**Metareview: Summary, Strengths And Weaknesses:**

This paper studies the problem of approximating a continuous system from discrete measurements with a finite data budget and provides a tradeoff between approximation and statistical error. However, reviewers raised concerns about the motivation and the simplicity of the theoretical model. The AC agrees and recommends rejection.